**Article** https://doi.org/10.1038/s41467-023-38571-w

# Functional specialization and interaction in the amygdala-hippocampus circuit during working memory processing

Jin Li [1,8], Dan Cao [1,8], Shan Yu[1,2], Xinyu Xiao[1,2], Lukas Imbach[3,4], Lennart Stieglitz [5], Johannes Sarnthein [5,6] ✉ & Tianzi Jiang [1,2,7] ✉

Both the hippocampus and amygdala are involved in working memory (WM) processing. However, their specific role in WM is still an open question. Here, we simultaneously recorded intracranial EEG from the amygdala and hippocampus of epilepsy patients while performing a WM task, and compared their representation patterns during the encoding and maintenance periods. By combining multivariate representational analysis and connectivity analyses with machine learning methods, our results revealed a functional specialization of the amygdala-hippocampal circuit: The mnemonic representations in the amygdala were highly distinct and decreased from encoding to maintenance. The hippocampal representations, however, were more similar across different items but remained stable in the absence of the stimulus. WM encoding and maintenance were associated with bidirectional information flow between the amygdala and the hippocampus in low-frequency bands (1–40 Hz). Furthermore, the decoding accuracy on WM load was higher by using representational features in the amygdala during encoding and in the hippocampus during maintenance, and by using information flow from the amygdala during encoding and that from the hippocampus during maintenance, respectively. Taken together, our study reveals that WM processing is associated with functional specialization and interaction within the amygdala-hippocampus circuit.

Various tasks in our everyday life require working memory (WM), for example in language processing, temporarily remembering the beginning of a phrase to make sense of the phrase as it closes. WM refers to a cognitive system storing information in an active and readily available state for a short period[1]. In humans, several brain areas are thought to be essential for WM[2]. Here we focus on two areas: the amygdala and the hippocampus. The amygdala is classically associated with emotional processing[3], while recent studies also showed that the amygdala has multidimensional response properties[4] and plays a role even in memorizing non-emotional stimulus material[5]. The hippocampus is typically studied for its role in long-term memories[6]. However, converging evidence points that the amygdala and the hippocampus are involved in WM[7], such as persistent neural firing[8–10] and elevated hippocampal activation[11,12] during WM processing. These studies suggested a general involvement of these areas in WM. However, their specific role in different WM phases has not been established.

[1]Brainnetome Center, Institute of Automation, Chinese Academy of Sciences, 100190 Beijing, China. [2]School of Artificial Intelligence, University of Chinese Academy of Sciences, 100049 Beijing, China. [3]Swiss Epilepsy Center, Klinik Lengg, Zurich, Switzerland. [4]Zurich Neuroscience Center, ETH and University of Zurich, 8057 Zurich, Switzerland. [5]Department of Neurosurgery, University Hospital Zurich, University of Zurich, 8091 Zurich, Switzerland. [6]Zurich Neuroscience Center, ETH Zurich, 8057 Zurich, Switzerland. [7]Research Center for Augmented Intelligence, Zhejiang Lab, 311100 Hangzhou, China. [8]These authors contributed equally: Jin Li, Dan Cao. ✉e-mail: johannes.sarnthein@usz.ch; jiangtz@nlpr.ia.ac.cn

Rather than looking at the mean level of activity, multivariate representational analysis methods allow the detection of specific patterns of activity and may be more informative about the representation of specific stimulus (see ref. [13] for a review). Previous studies identified two properties related to memory performance, the representational dissimilarity among different stimulus[14] and the representational stability between different memory periods[15]. The amygdala is known as a detector of goal-related stimuli[16] and receives major projections from the anterior temporal lobe[17] that convey highly processed object information[18]. On the other hand, the hippocampus is crucial for memory consolidation[19]. For instance, recent studies suggested notable overlap in representational patterns between the encoding and the post-encoding period in the hippocampus[15,20]. Therefore, we hypothesized a functional specialization in the amygdala and the hippocampus in WM encoding and maintenance periods. However, no study has yet simultaneously tracked amygdala and hippocampal representations in humans. Whether the perceptual representations differed in the amygdala and the hippocampus or whether they changed from the encoding to the maintenance phase remained unclear.

WM relies on processing in functionally interconnected brain areas[2]. But how do the amygdala and the hippocampus work together to support WM? The inter-regional communications have started to be addressed. At the anatomical level, tract tracing studies have uncovered structural connections between the amygdala and the hippocampus[21]. At the functional level, electrical stimulation of the hippocampus can induce synaptic plasticity in the amygdala in rodent studies[22] and human studies indicated that stimulation of the amygdala led to increased power in the hippocampus[5]. The structural and electrophysiological evidence suggests inter-regional communication. However, studies of inter-regional communication between the amygdala and hippocampus primarily focused on emotional memory[20,23], not WM processing with non-emotional contents.

Here, we investigated how the two areas interact and transfer WM related information during WM. To address these issues, we recorded iEEG simultaneously from the amygdala and the hippocampus in human epilepsy patients while they performed a WM task. By combining the high temporal resolution of human iEEG recordings with a variety of approaches including representational similarity analysis (Fig. 1a, b), information flow analysis (Fig. 1c) and neural pattern classification analysis (Fig. 1d), the current study examined the aspects of memory representations in the amygdala and the hippocampus and their interactions that contribute to WM. We found that the amygdala forms distinct mnemonic representations during encoding while the hippocampus keeps stable representations from encoding to maintenance. Next, we observed enhanced inter-regional information transfer during both encoding and maintenance. Finally, the functional specialization and interaction patterns were predictive of WM load.

## Results

### Task, behavior, and recording channels

Fourteen patients with drug-resistant epilepsy (7 females) (Table 1) performed a modified Sternberg WM task (65 total sessions from 14 participants) during an invasive presurgical evaluation. In this task, the items were presented simultaneously rather than sequentially, thus separating the encoding period from the maintenance period. In each trial, the participant was asked to memorize a set of 4, 6, or 8 letters presented for 2 s (encoding). The number of letters was thus specific for the memory load. After a delay (maintenance) period of 3 s, a probe letter was presented and the participant responded whether the probe letter was identical to one of the letters held in memory (retrieval) (Fig. 2a). The average accuracy was 91.9% ± 3.2% (range 86.1% − 97.6%). The mean response time was faster for correct than incorrect trials (1.44 ± 0.36 versus 1.95 ± 0.66 sec, paired *t-test*: $t(13) = -4.15$, $p = 0.0011$). Hence, the participants performed well in the task. We also

tested the effect of set size (WM load) on the participants' response accuracy. We found that the accuracy of WM decreased from load 4 (mean ± S.D.: 98.04% ± 1.91%) to load 6 (90.78% ± 5.36%) and 8 (85.36% ± 5.89%) (repeated-measures analysis of variance (ANOVA), $F(2,26) = 42.71$, $p < 0.001$, Fig. 2b). This finding indicates that the behavioral performance was modulated by WM load, which is in line with previous study that the factor of load had a significant impact on working memory performance[10].

Local field potentials (LFPs) were recorded simultaneously from depth electrodes implanted in the amygdala and hippocampus (Fig. 2c). In total from all participants, we recorded from 92 channels in the hippocampus and 50 channels in the amygdala (Table 1, see the details in "Methods").

### Functional specification: distinct representation within the amygdala during encoding

We applied multivariate analysis to investigate how WM information is represented across neural activation patterns within the amygdala and the hippocampus. We performed a series of representational similarity analyses to investigate two representational properties crucial for memory performance, i.e., the distinctiveness and the stability of neural representations[24].

In the first analysis, we examined whether there was representational distinctiveness during WM encoding within the two regions. To address this question, we first performed an encoding-encoding dissimilarity (EED) analysis (see "Methods"). Neural activity on 1–40 Hz frequencies was included in representational analyses. We chose this frequency range for the following two reasons. First, both the amygdala and the hippocampus showed elevated activity in the low-frequency range 1–40 Hz (Fig. S1a). Second, the averaged z-scored power on 1–40 Hz was significantly above zero at most time points during WM processing, while only a few time points showed significantly elevated activity for the averaged z-scored power on 40–100 Hz (Fig. S1b). These findings are in line with our previous study[25].

We then correlated the representational patterns from every two trials across channels and frequencies (1 to 40 Hz in steps of 1 Hz) in consecutive overlapping time windows of 100 ms (step width 10 ms Fig. 2d). The dissimilarity (1 − similarity) of the representational patterns was averaged across all trial-pairs, resulting in a temporal EED map across all participants for the amygdala (Fig. 2e left) and the hippocampus (Fig. 2e right) separately. Next, we compared the EED map between the amygdala and the hippocampus using cluster-based permutation tests. Across all encoding time windows, a cluster with higher EED values in the amygdala than in the hippocampus appeared (outlined in black in Fig. 2f, cluster-based permutation, $p = 0.007$); no cluster with higher EED values in the hippocampus than in the amygdala was observed. Further, we averaged the EED values in significant clusters for all participants and compared them for the amygdala and the hippocampus. This revealed higher EED values in the amygdala than in the hippocampus across participants (paired *t-test*, $p = 0.027$, $t(13) = 2.49$, Fig. 2g). As the letter strings were different across trials, these findings indicated that the activity patterns for different items had a larger distance among each other within the amygdala, whereas the hippocampus showed overlapping representations across different items with reduced neural dissimilarity.

We could think of two possible explanations for the higher EED in the amygdala, one is the specific representation of distinct items. Alternatively, it could also be due to higher activity fluctuation across trials in the amygdala. To *test* this explanation, we calculated the variability (standard error) of the averaged z-scored power at 1–40 Hz during the encoding period across trials, in the amygdala and the hippocampus for each participant, respectively. Then we compared them using a paired *t-test*. Results showed no difference between the variability in the amygdala and that in the hippocampus (paired *t-test*,

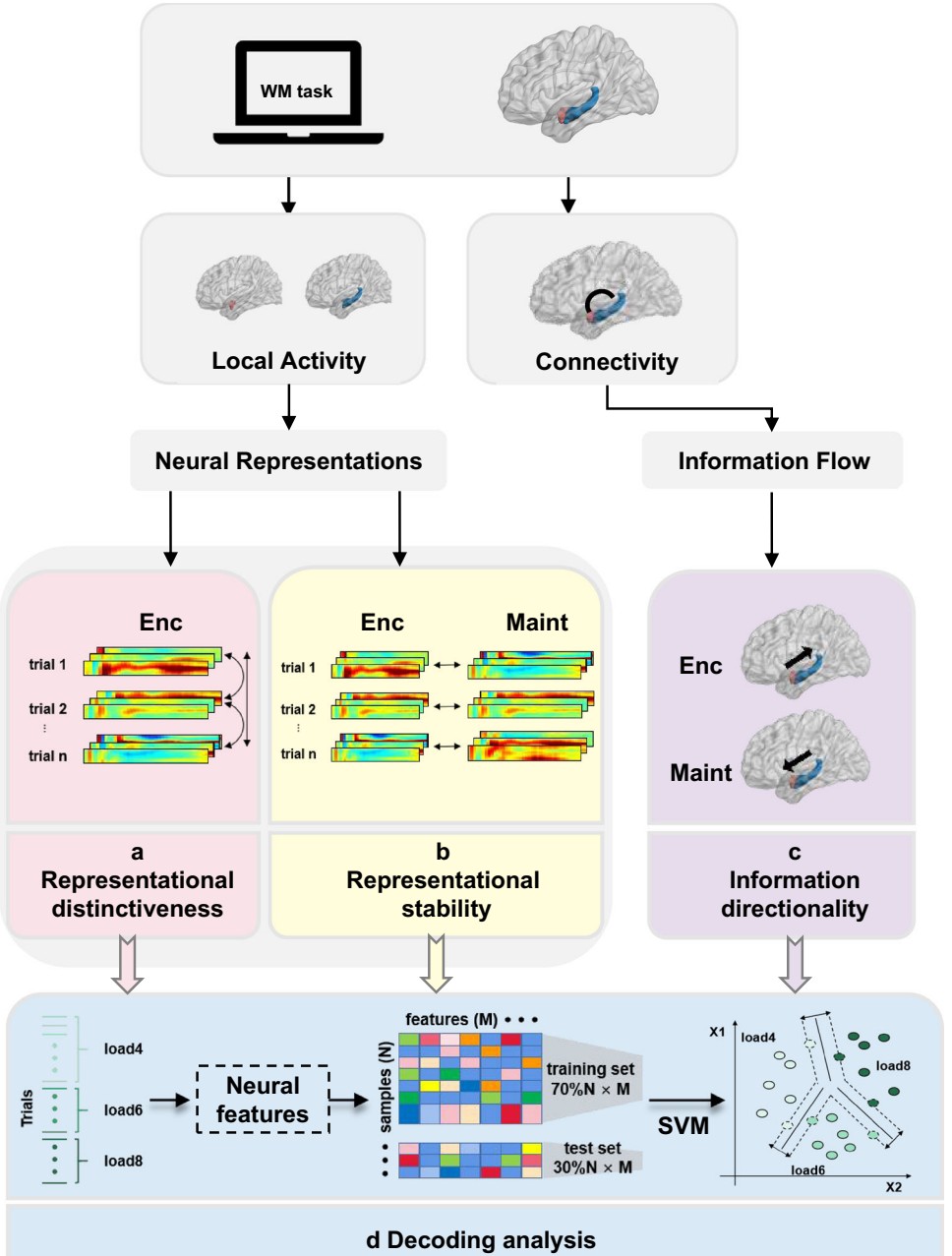

**Fig. 1 | Study framework.** To investigate the function of the amygdala and hippocampus in WM processing, we performed a series of analyses. **a** First, we calculated the encoding-encoding representational dissimilarity (EED) between trial pairs; **b** then, the encoding-maintenance similarity (EMS) in the same trial was computed; **c** next, we measured the directional information flow between the amygdala and the hippocampus; **d** decoding analysis: we used machine-learning analyses to investigate whether WM load (load 4, 6, and 8) could be predicted by encoding-encoding dissimilarity (EED), encoding-maintenance similarity (EMS), or phase slope index (PSI) features. The brain figure was visualized by BrainNet Viewer toolbox (www.nitrc.org/projects/bnv/) Xia et al.[43].

$p = 0.94$, Fig. S2a). Next, we compared the activity variability at each time point in the encoding period between the amygdala and the hippocampus. Again, the activity variability showed no significant difference at each time point throughout the encoding period (cluster-based permutation *test*, $p > 0.05$, Fig. S2b). These findings indicated that less similarity across encoding items in the amygdala could not be explained by differences in power variability.

**Functional specification: stable representation within hippocampus during maintenance**
We next examined whether and how stable representational structures were maintained in the absence of stimuli within the amygdala and the hippocampus. Representational stability was indexed by memory

reinstatement, an approach borrowed from the long-term memory literature[26]. Memory stability was quantified as the correlation between patterns of oscillatory power across channels and frequencies (1 to 40 Hz in steps of 1 Hz) within consecutive overlapping time windows of 100 ms (step width 10 ms) for each combination of the encoding-maintenance time bins in the same trial. The correlation matrix was then averaged across trials, resulting in a temporal map of encoding-maintenance similarity (EMS) for the amygdala and the hippocampus (Fig. 3a). The EMS values were higher in the hippocampus than in the amygdala for every encoding-maintenance time pair in the EMS map (Fig. 3b). And, the averaged EMS values in the hippocampus was higher than that in the amygdala across participants (paired *t-test*, $p = 0.0049$, $t(13) = 3.38$, Fig. 3c). These findings indicated

**Table 1 | Participant characteristics**

| Participant | Recording sites (amygdala) | Recording sites (hippocampus) | Retrieval accuracy (%) | RT (s) for correct trials (mean ± std) |
|---|---|---|---|---|
| 1 | 2 | 6 | 92.5 | 1.26 ± 0.63 |
| 2 | 4 | 8 | 86.4 | 1.28 ± 0.65 |
| 3 | 2 | 6 | 93.2 | 1.10 ± 0.34 |
| 4 | 4 | 8 | 94.9 | 1.48 ± 0.63 |
| 5 | 2 | 4 | 91.3 | 1.41 ± 0.90 |
| 6 | 4 | 8 | 94.3 | 1.47 ± 0.59 |
| 7 | 4 | 8 | 94.9 | 1.44 ± 0.57 |
| 8 | 4 | 7 | 90.0 | 1.41 ± 0.51 |
| 9 | 4 | 8 | 92.8 | 1.24 ± 0.33 |
| 10 | 4 | 6 | 93.0 | 1.64 ± 0.73 |
| 11 | 4 | 6 | 91.4 | 1.30 ± 0.58 |
| 12 | 4 | 5 | 89.5 | 1.49 ± 0.77 |
| 13 | 4 | 4 | 97.6 | 1.05 ± 0.28 |
| 14 | 4 | 8 | 86.1 | 2.58 ± 1.22 |

that the hippocampus retained WM information in a more stable representation during maintenance than the amygdala.

We next exclude the possibility that the inter-regional differences were due to the presence of visual information during encoding and its absence during maintenance. To this end, we calculated the difference of power between the two periods ((maintenance−encoding)/encoding) to index the processing of visual information in the amygdala and the hippocampus and made comparison between regions using paired $t$-tests. No difference was found between the two regions (paired $t$-test, $p = 0.09$, Fig. S2e). Among all 14 participants, the relative difference of power was higher within the amygdala in 7 participants and higher within the hippocampus in 7 participants. Further, we also extracted this relative difference of power to decode the WM load. As shown in Fig. S2f, the relative difference of power could not decode the WM load regardless of using features within the amygdala (32.17% ± 10.79%) or within the hippocampus (32.58% ± 8.13%; permutation $test$ against scrambled data, $p > 0.05$). No difference of decoding accuracy was found between the two regions (paired $t$-test, $p = 0.76$).

**Functional interaction: bidirectional information transfer within the amygdala-hippocampal circuit during WM encoding and maintenance**

In addition to functional specialization, functional interaction between brain regions is considered to be another important mechanism in WM[25]. To $test$ whether the amygdala and hippocampus work independently or interactively in WM, we investigated the directionality of information transfer between the amygdala and hippocampus. We used Phase Slope Index (PSI) to quantify the directional connectivity between the amygdala and the hippocampus[27]. PSI quantifies phase difference as a function of frequency, with a positive value indicating that the signal from the first structure is leading the signal from the second structure. PSI was computed for the data segments during encoding and maintenance for all correct trials from 1 to 40 Hz and tested for significance of directional effects via a nonparametric permutation procedure (see "Methods" for details). The directional effects were averaged across channel pairs and participants, yielding a $z$-score that indicates the information flow between the amygdala and the hippocampus during encoding and maintenance, respectively, as previous studies did[28]. The encoding period was characterized by unidirectional hippocampus-to-amygdala connectivity across 1−11 Hz (threshold $z > 1.96$, $p < 0.05$), and unidirectional amygdala-to-hippocampus connectivity across 13−27 Hz and 36−40 Hz (threshold $z < −1.96$, $p < 0.05$, Fig. 4a). The maintenance period was characterized

by unidirectional hippocampus-to-amygdala connectivity across 1−18 Hz (threshold $z > 1.96$, $p < 0.05$), and unidirectional amygdala-to-hippocampus connectivity across 23−30 Hz and 33−34 Hz (threshold $z < −1.96$, $p < 0.05$, Fig. 4b). These results indicated a frequency-specific directional connectivity in the amygdala-hippocampal circuit involved in WM processing. Specifically, the direction of influence differed across frequency band, with theta/alpha-driven unidirectional influence from the hippocampus and beta-driven influence from the amygdala. Besides, frequency bands for both directions varied between the encoding and the maintenance period. This resulted in a wider frequency band range showing amygdala leads influence than the opposite direction in the encoding period, and a wider frequency with hippocampal leads influence in the maintenance period. These findings are consistent with findings in the representational analyses showing contribution of the amygdala to WM encoding and the hippocampus to WM maintenance.

**Functional specialization and interaction within the amygdala-hippocampal circuit predicted WM load**

To address whether the EED patterns within the amygdala or the hippocampus could predict WM load, we developed an approach that uses the EED patterns in the amygdala as well as the hippocampus to predict the WM load (load 4, 6 or 8) with a linear support vector machine (SVM)[29] classifier. For each load, the EED patterns at trial-pairs level for all participants were pooled as the data used in the classification. We randomly extracted 70% of the data from each load and pooled them across all loads to train the SVM classifier. We tested the classifier in the remaining data to obtain the decoding accuracy as our performance measure. The procedure was repeated 100 times for cross-validations (see details in "Methods"), and the accuracy of the classifier was averaged across these 100 cross-validations to measure its performance. The significance of the difference in decoding accuracy between the amygdala and hippocampus was assessed using a nonparametric permutation test. Specifically, we compared the actual difference with a null distribution obtained from scrambled labels. As shown in Fig. 2h, the decoding accuracy from the amygdala EED pattern (33.54% ± 1.31%) was significantly higher than the hippocampus EED pattern (32.96% ± 1.16%; permutation $test$: $p = 0.01$). We also performed analogical decoding analysis using the EMS patterns within the amygdala or the hippocampus, as described in the EED patterns. Results (Fig. 3d) showed that the decoding accuracy (35.33% ± 1.70%) from the hippocampus EMS pattern was significantly higher than the amygdala EMS pattern (34.23% ± 1.50%; permutation $test$: $p < 0.001$).

We next asked whether the inter-regional interaction between the amygdala and the hippocampus could predict WM load. Based on previous observations, we separately extracted the directional connectivity from the hippocampus leads and the amygdala leads on the $z$-scored PSI at each channel pair for each participant. Then, for both directions, the directional connectivity at channel-pairs level for all participants were pooled as the data for each load classification (see details in "Methods"). Similar decoding analyses were performed for each direction during encoding and maintenance. As presented in Fig. 4c, during encoding, the decoding accuracy using the features from the amygdala leads (42.94% ± 3.22%) was significantly higher than the opposite direction (40.85% ± 3.08%; permutation $test$: $p < 0.001$). During maintenance (Fig. 4d) the decoding accuracy using the features from the hippocampus leads (45.62% ± 3.57%) was higher than the opposite direction (43.75% ± 3.21%; permutation $test$: $p < 0.001$).

In addition, we made a random effects analysis to directly compare the EED patterns between WM load. We treated the regions (amygdala/hippocampus) and WM load (low (set4)/high (set6/8)) as fixed factors, the participants as random factor, and the extracted EED values as the dependent variables. We observed higher EED values for high-load trials versus low-load trials regardless of the regions, although the load effect did not reach significance (mixed-effect

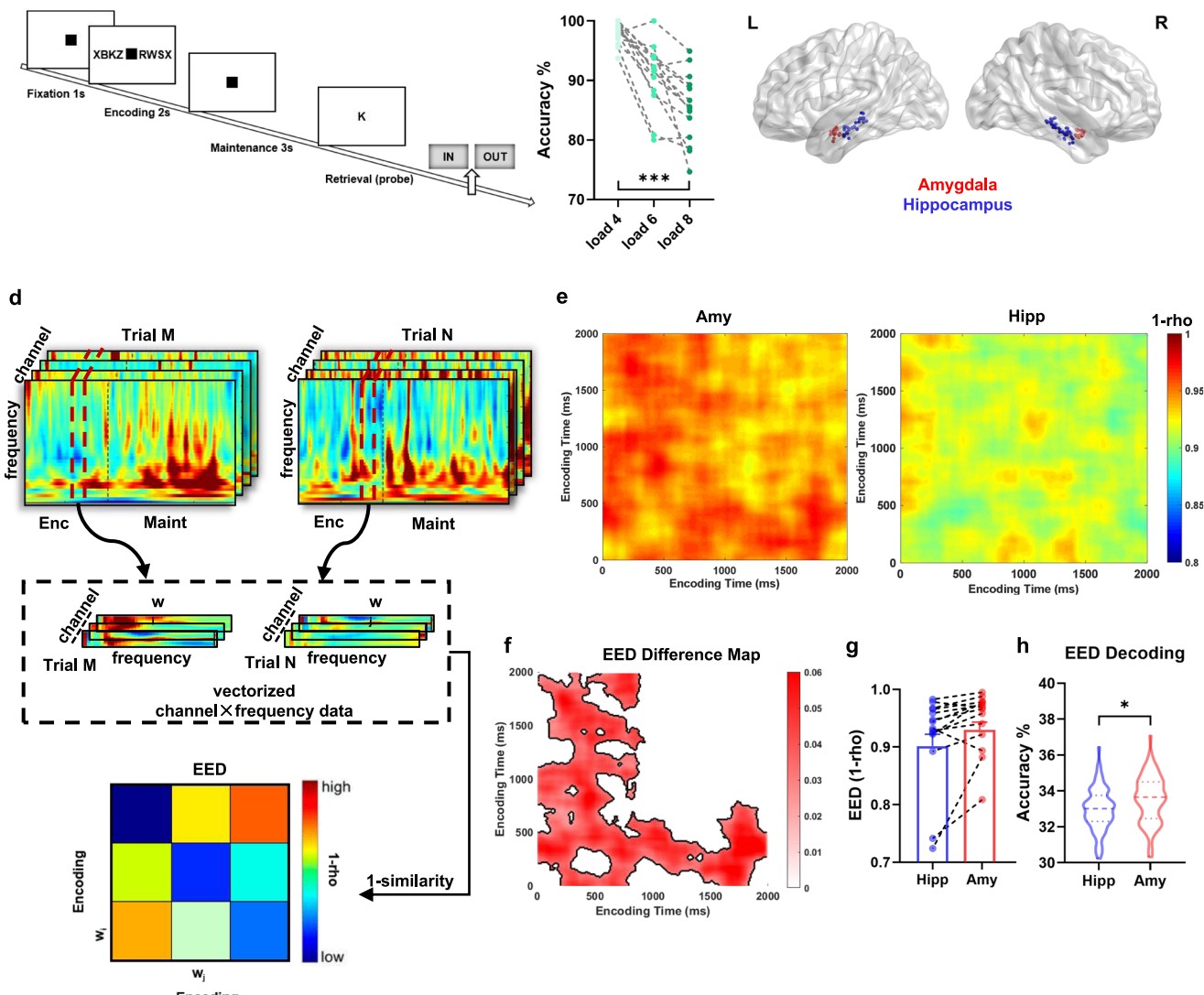

**Fig. 2 | Working memory task, recording sites, representational dissimilarity analysis and decoding analysis during encoding. a** In each trial, a set of consonants was presented (encoding 2 s) followed by a delay (maintenance 3 s). Then a probe letter was shown and the participants indicated whether the probe was in the initial set of consonants (retrieval). **b** Accuracy decreased from load 4 to load 6 and 8 (repeated-measures ANOVA, $p < 0.001$, $F(2,26) = 42.71$). A line connects the data from one participant ($n = 14$). ***$p < 0.001$. **c** Channel location across participants in MNI152 space. Recording regions are indicated by different colors (red, amygdala; blue, hippocampus). The brain figure was visualized by BrainNet Viewer toolbox (www.nitrc.org/projects/bnv/) Xia et al.[43]. **d** Schematic of encoding-encoding dissimilarity (EED) analysis. Warmer color denotes higher dissimilarity and cooler color means higher similarity. **e** Averaged EED map during the encoding period across all participants, in the amygdala (left column) and the hippocampus (right column). **f** EED Difference map obtained by subtracting the hippocampus EED from the amygdala EED reveals a significant cluster ($p < 0.05$, two-sided cluster-based permutation *test*, outlined in black), indicating that the amygdala represents the working memory information specifically during the encoding period (dark red area, higher EED in the amygdala than the hippocampus). White areas indicate that there was no significant difference ($p > 0.05$) between the hippocampus and amygdala. **g** EED values averaged over the significant cluster in **b** was extracted within the hippocampus (blue, mean ± s.e.m.) and amygdala (red, mean ± s.e.m.) for each participant, respectively. 12 of 14 participants showed higher EED values within the amygdala than within the hippocampus. **h** Decoding accuracy using the EED patterns within the amygdala (red) is higher than in the hippocampus (blue) from all cross-validations ($n = 100$; two-sided permutation *test*: $p = 0.01$). Dotted lines indicate the median. Broken lines above and below denote the quartiles. *$p < 0.05$. Source data are provided as a Source data file.

model: $p = 0.14$; Fig. S2c). Similar comparison was also made between WM load as described in EED patterns. As shown in Fig. S2d, lower EMS values with high-load trials were obtained relative to the low-load trials regardless of the regions, although the load effect did not reach significance (mixed-effect model: $p = 0.25$).

Taken together, our collective results that WM load can be predicted by representational features in the amygdala during encoding and in the hippocampus during maintenance, and by information flow from the amygdala during encoding and that from the hippocampus during maintenance indicated that the amygdala contributed to WM

encoding and the hippocampus participated in WM maintenance in a load-sensitive manner.

## Success effect for high load conditions
In addition, we examined whether the neural representation and directional connectivity varied as a function of behavioral success, as the participants' performance in the high load conditions (combined load 6 and 8) was not at ceiling (mean performance over all participants 87%). First, we separately computed the EED and EMS for the correct and incorrect trials in the high load conditions. We then

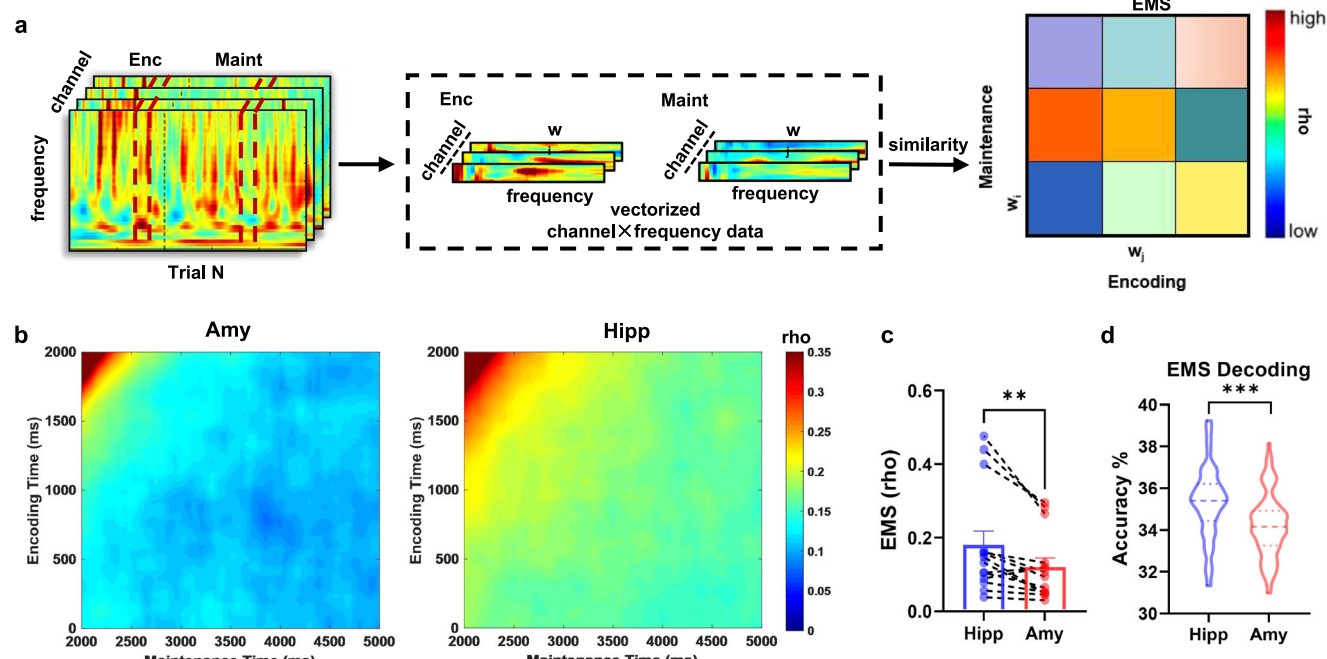

**Fig. 3 | Representational stability within the amygdala and the hippocampus and decoding analysis. a** Schematic of encoding-maintenance similarity (EMS) analysis. Warmer color denotes higher similarity and cooler color means less similarity. **b** Grand average EMS map across all participants in the amygdala and the hippocampus. **c** The average EMS was higher in the hippocampus (blue, mean ± s.e.m.) than in the amygdala (red, mean ± s.e.m.; two-sided paired *t*-test: *p* = 0.0049, $t(13) = 3.381$). Each dot represents one participant ($n = 14$). **$p < 0.01$. **d** The decoding accuracy by using the EMS patterns in the hippocampus (blue) is higher than in the amygdala (red) from all cross-validations ($n = 100$; two-sided permutation *test*: $p < 0.001$). Dotted lines indicate the median. Broken lines above and below denote the quartiles. ***$p < 0.001$. Source data are provided as a Source data file.

performed two 2 (Performance: correct vs. incorrect) × 2 (Region: amygdala vs. hippocampus) repeated-measures ANOVAs, one with the EED value and one with the EMS value as the dependent variable. As shown in Fig. S3a, the EED value of the correct trials were significantly greater than incorrect trials ($p = 0.026$, $F(1,13) = 6.28$). This indicated that the activity patterns for correct trials showed a larger distance among different trials, whereas incorrect trials showed overlapping representations across different items with reduced neural dissimilarity. The EMS showed no significant difference between the correct trials and incorrect trials ($p = 0.17$). Regarding to the PSI, for each participant, we separately extracted the PSI from the hippocampus leads and the amygdala leads for the correct and the incorrect trials. We made comparisons between regions and performance by using repeated-measures ANOVAs. During encoding, we found a significant interaction effect ($p = 0.019$, $F(1,13) = 7.23$). Further analysis showed that the amygdala leads connectivity was significantly larger in the correct trials than the incorrect trials ($p = 0.042$) and no difference between correct and incorrect trials was found from the opposite direction ($p = 0.21$, Fig. S3b). During maintenance, a significant interaction effect ($p = 0.025$, $F(1,13) = 6.43$) was also found. Further analysis showed that the hippocampus leads connectivity was significantly larger in the correct trials than the incorrect trials ($p = 0.034$) and no difference was found between correct and incorrect trials from the opposite direction ($p = 0.14$, Fig. S3c). This again indicated that the information flow driven by the amygdala during encoding and that driven by the hippocampus during maintenance contributed to WM.

Next, we also applied the EED/EMS/PSI patterns within the amygdala and the hippocampus to decode the performance (correct or incorrect) in the high load conditions. Using the SVM classifier as described before, we found that the decoding accuracy by using the EED pattern within the amygdala (61.50% ± 13.73%) was higher than the hippocampus (50.25% ± 10.66%; permutation *test*: $p < 0.001$; Fig. S3d). The decoding accuracy by using the EMS pattern within the

hippocampus (59.63% ± 12.29%) was higher than the amygdala (56.13% ± 10.88%), although the difference did not reach significance (permutation test: $p = 0.062$; Fig. S3e). Besides, during encoding, decoding accuracy by using the "amygdala leads" PSI (59.50% ± 17.69%) was higher than the "hippocampus leads" PSI (53.88% ± 14.40%; permutation *test*: $p = 0.005$; Fig. S3f); and during maintenance, the decoding accuracy by using the "hippocampus leads" PSI (61.75% ± 11.35%) was higher than that the "amygdala leads" PSI (57.88% ± 14.40%; permutation *test*: $p = 0.025$; Fig. S3g). These results indicate that the contribution of EED within the amygdala and amygdala leads directional connectivity on WM performance during encoding, and the contribution of EMS within the hippocampus and hippocampus leads directional connectivity on WM performance during maintenance.

## Discussion
By analyzing neural representations and inter-regional information flow, we found that (1) the amygdala was involved in forming distinct memory representations for different items during encoding; (2) the hippocampus retained the representation of WM information in the absence of a stimulus on the screen; (3) WM encoding and maintenance were associated with enhanced bidirectional inter-regional information flow; and (4) WM load could be decoded by representational features in the amygdala during encoding and in the hippocampus during maintenance, and by information flow from the amygdala during encoding and that from the hippocampus during maintenance.

While the amygdala is known to serve emotion processing, its contribution to WM encoding of non-emotional items may seem surprising. However, a significant body of evidence has indirectly implicated a role of the amygdala in memory encoding. First, a recent human study reported that the amygdala has a higher proportion of concept cells with specific responses to preferred stimulus than the hippocampus[9]. Other studies suggested that the amygdala plays

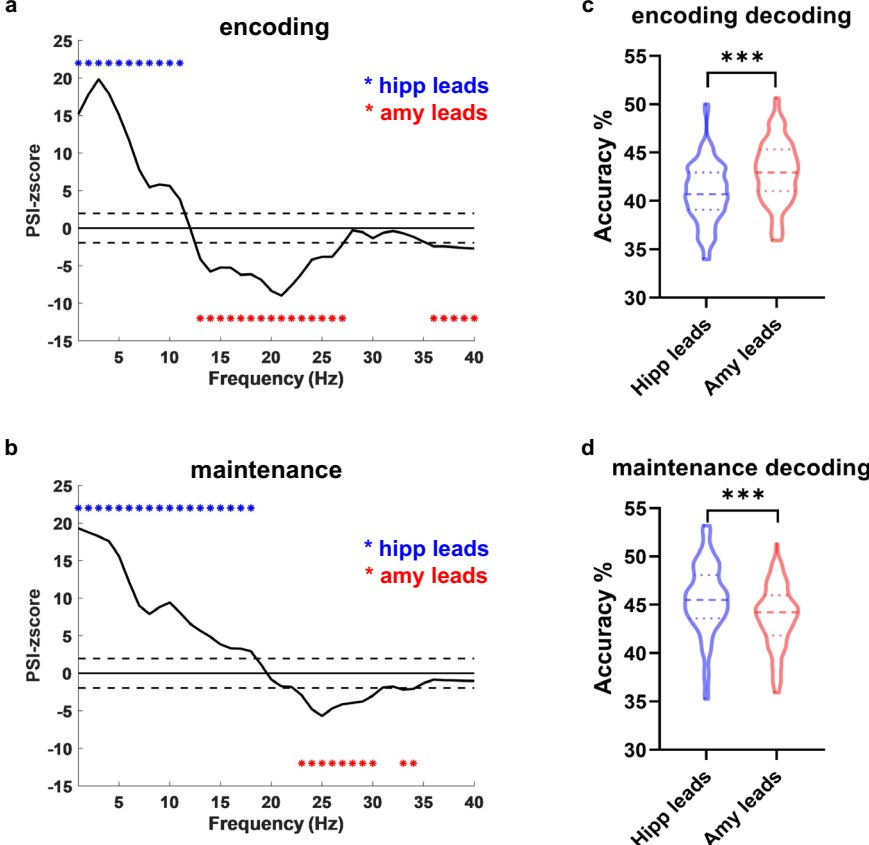

**Fig. 4 | Directional information flow between the hippocampus and the amygdala during encoding and maintenance. a** The z-scored phase slope index (PSI) across 1–40 Hz during encoding. Asterisks in blue denote significant PSI from the hippocampus to the amygdala and these in red denote the opposite direction (significance was thresholded at |z| > 1.96). **b** The z-scored PSI across 1–40 Hz during maintenance. **c** Decoding accuracy of WM load by using PSI features of the amygdala leads connectivity (red) was higher than those of the hippocampus leads connectivity (blue) from all cross-validations (n = 100) during encoding (two-sided permutation *test*: p < 0.001). Dotted lines indicate the median. Broken lines above and below denote the quartiles. ***p < 0.001. **d** Decoding accuracy of WM load by using PSI features of the hippocampus leads connectivity (blue) was higher than those of the amygdala leads connectivity (red) from all cross-validations (n = 100) during maintenance (two-sided permutation *test*: p < 0.001). ***p < 0.001. Source data are provided as a Source data file.

a stimulus specific role in novelty detection[30,31] and encodes state-dependent exploratory behavior[32]. Both functions require distinct memory representations among different encoding items. Our finding is also consistent with recent literature highlighting a broader function for the amygdala, including processing of sensory, memory, valence, etc[4]. Taken together, our study provides evidence that the amygdala might contribute to WM by encoding non-overlapping memories with distinct representations.

The hippocampal representations showed less distinctiveness but more stability between the encoding and the maintenance period. The observed lower EED corroborates a recent view that the hippocampus supports more overlapping representations that hold across experiences[33]. The higher EMS in the hippocampus is in line with our previous work revealing a higher proportion of maintenance cells in the hippocampus than in the amygdala[10]. Studies in long term memory also suggested that memory consolidation in the hippocampus may start early, even at the end of encoding[20], suggesting that during this post-encoding period, memory representations are maintained in the hippocampus as memory engrams, so that they can be used to recover information later. Taken together, we infer that the hippocampus may contribute to WM maintenance by keeping representations stable in the absence of the stimulus.

In addition to the functional specialization, we found bidirectional inter-regional interaction between the amygdala and the hippocampus during WM encoding and maintenance. The inter-regional communication is consistent with rodent[34] and monkey[35] studies that

report anatomical connection by tracing techniques, and with rodent studies that find synaptic plasticity in the amygdala induced by electrical stimulation to the hippocampus[22] and vice versa[36]. Human iEEG studies found inter-regional functional connectivity during emotional information[37] and emotional memory[23] processing. In our current study, we extended the functional interaction in the amygdala-hippocampal circuit to WM processing even in the absence of explicit emotional content. We further found that the information flow from the amygdala contributes to WM encoding and that the information flow from the hippocampus contributes to WM maintenance in a load-sensitive manner. This is in line with our previous study with the same task in which the inflow to the hippocampus during the encoding period transfers external sensory information from primary auditory cortex, while the outflow from the hippocampus during the maintenance period transfers memory information as memory replay[38].

There are two major views on the brain's cognitive function. The first view emphasizes that modularity supports functional specialization[39]. Our results support this view by showing more distinct representational patterns in the amygdala during encoding and a higher encoding-maintenance representational similarity in the hippocampus. The second view emphasizes distributive processing where the brain is highly interactive and its regions are functionally interconnected[40]. Our findings support this view by showing inter-regional information flow. During encoding, the amygdala forms highly distinct representations for different encoding items and

conveys this perceptual information to the hippocampus. During maintenance, the hippocampus can well retain the encoded representational patterns and transfers this memory information back to the amygdala. Interestingly, we also observed that the representational features in specific region/direction during specific WM period could predict WM load. Thus, our finding of modular processing in the amygdala and the hippocampus at different WM periods and the distributed processing in their interactions propose a mechanism how the amygdala-hippocampus circuit supports WM processing.

In summary, our results demonstrated functional specialization between the amygdala and hippocampus and their inter-regional communication in WM processing. We provide a mechanistic explanation of how neuronal activity patterns in the two structures differentially contribute and orchestrate to support working memory.

## Methods

### Participants

Data were obtained from epilepsy patients undergoing intracranial EEG monitoring at the Swiss Epilepsy Center, Klinik Lengg, Switzerland, to localize epileptic foci for potential surgical resection. Intracranial depth electrodes (1.3 mm diameter, 8 contacts of 1.6 mm length, 5 mm spacing; Ad-Tech, Racine, WI, www.adtechmedical.com) were stereotactically implanted. The electrode placements were guided exclusively by clinical needs. We included all patients that accepted to participate in the study and that had electrodes implanted in the amygdala and hippocampus in the same hemisphere. Before *testing*, all participants provided written informed consent for the study, which had been approved by the relevant institutional ethics review board (Kantonale Ethikkommission Zürich, PB 2016–02055). In total, 14 participants (mean ± SD [range]: 34.5 ± 12.6;[18–56] 7 females) participated in this study. There were no seizures recorded during any of the epochs, and any epochs with interictal epileptiform activity were excluded from analysis.

### Task

We used a modified Sternberg task in which the encoding of memory content, maintenance, and recall were temporally separated (Fig. 2a). Each trial started with a fixation period (1 s) followed by the presentation of the stimulus in the encoding period (2 s). In each trial, the participant was asked to memorize a set of 4, 6, or 8 letters presented for 2 s (encoding). The number of letters was thus specific for the memory load. After the encoding period, the stimulus was replaced by a fixation square during the maintenance period (3 s). Finally, a probe letter appeared and the participants responded with a button press ("IN" or "OUT") to indicate whether the probe was part of the stimulus letter set. The participants performed 50 trials per session, which lasted approximately 10 min. Some participants performed up to seven sessions of the task.

### Channel localization

The channels were localized using postimplantation computed tomography (CT) scans and postimplantation structural T1-weighted MRI scans. For each patient, the CT scan was co-registered to the postimplantation scan, as implemented in FieldTrip[41,42]. The channels were visually marked on the coregistered CT-MR images. Channel positions were verified by the neurosurgeon (L.S.) after merging pre-operative MRI with postimplantation CT images of each individual patient in the plane along the electrode (iPlan Stereotaxy 3.0, Brainlab, München, Germany). Channel locations in native space for each patient were projected to MNI space and are shown in Fig. 2c, which was visualized by the BrainNet Viewer[43]. The final dataset contained 94 channels in the hippocampus and 50 channels in the amygdala across all patients. There were 6.7 ± 1.4 (range 4–8) channels per patient in the hippocampus and 3.6 ± 0.9 (range 2–4) channels per participant in the amygdala.

### Channel selection

Each participant had 0–1 electrode targeting the anterior and posterior hippocampus and the amygdala per hemisphere. Targeted regions and hemispheres varied across participants for clinical reasons and included the hippocampus in the left ($n = 13$) and right ($n = 14$) hemispheres and the amygdala in the left ($n = 13$) and right ($n = 11$) hemispheres. We selected the two most medial channels on each electrode targeting the hippocampus or the amygdala, as was done in previous studies[26,44]. This procedure was used to minimize inter-individual variability, which would be higher if different numbers of channels would have been selected across participants. The final number of selected channels in each region for each participant is listed in Table 1. We included only ipsilateral channel pairs in the analysis.

### Data acquisition and preprocessing

Intracranial data were acquired against a common intracranial reference using a Neuralynx ATLAS recording system, sampled at 4 kHz, and analog-filtered above 0.5 Hz. After data acquisition, neural recordings were down sampled to 1 kHz and band-pass filtered between 1 to 200 Hz using the zero-phase delay finite impulse response (FIR) filter with Hamming window. Line noise harmonics were removed using a discrete Fourier transform. The filtered data were manually inspected to mark any channels containing epileptiform activity or artifacts for exclusion. The data were then re-referenced to the average of the signal over all the clean channels[12]. We then segmented the preprocessed data into event-related epochs; 1 s fixation period, 2 s encoding period, 3 s maintenance period, and 2 s retrieval period. We rejected trials with artifacts by visual inspection (53/3250 or 1.6% of all trials). Subsequent analyses were performed on correct trials. We performed preprocessing routines with the FieldTrip[41], EEGLAB[45] toolboxes and custom scripts in MATLAB version R2018b (the MathWorks, Natick, MA, USA).

### Time-frequency analysis

Time-frequency power was computed for each selected channel at each trial within the hippocampus as well as the amygdala in each participant. For each trial at each channel, we convolved the signal with complex-valued Morlet wavelets (6 cycles) to obtain power information at each frequency from 1 to 100 Hz in 1 Hz steps with a time resolution of 1 ms[15]. The task-induced power was analyzed per trial using a statistical bootstrapping procedure as was done in previous studies[12,46]. Briefly, for each channel and frequency, a null distribution was created by randomly selecting and averaging several data points in the baseline power (500 ms pretrial) 1,000 times, the raw power for each time point during task was then *z*-scored by comparing it to the null distribution to generate the *z*-scored power. Both the encoding and maintenance of WM information elicited *z*-scored power changes across 1–40 Hz frequency range (Fig. S1). Therefore, the *z*-scored power in this frequency range was used as the feature for the subsequent analyses.

### Representational dissimilarity analysis

A sliding time window approach was applied to calculate the representational dissimilarity in a 100 ms sliding time window (step width 10 ms). The *z*-scored power was first averaged across the time points within each sliding window for each trial. The generated *z*-scored power of all channels and frequencies (1–40 Hz) within each time window were then vectorized. The Spearman's correlation between the features of the two trials was calculated and Fisher *z*-transformed. The generated values were subtracted from 1 and then averaged across trial pairs to index the dissimilarity among trials in the given time window pair. After these steps, we got the encoding-encoding dissimilarity (EED) map across all time windows during encoding. The analysis procedure is presented in Fig. 2d. We then compared the EED map in the amygdala with the hippocampus at the group level by using a cluster-based permutation *test*[47]. We also extracted the average EED

values in the significant cluster in the two regions for each participant and then contrasted them via paired *t*-tests at the group level.

### Representational similarity between encoding and maintenance

Next, we calculated the representation similarity between encoding and maintenance (EMS) periods within the same trials for the correct trials. First, we built representational patterns based on distributed oscillatory power (1–40 Hz) across all channels for each participant between all pairs of time windows (one from encoding and the other from maintenance), resulting in EMS maps within the same trials between all encoding-maintenance time window pairs, which were then averaged across trials. Figure 3a describes the detailed analysis procedure. Next, we contrasted the EMS maps for the amygdala and the hippocampus by using a cluster-based permutation *test*. Since we found significant contrast in every encoding-maintenance time pair in the EMS map, we then extracted the average EMS values in the whole map for each participant from the two regions and contrasted them via paired *t*-tests at the group level.

### Phase slope index analysis

PSI estimates whether the slope of the phase differences between A-B signal pairs is consistent across several adjacent frequency bins, in which positive PSI indicates that signal A leads signal B, and negative PSI indicates the reverse[46]. In this study, the data segments during encoding and maintenance were zero padded and multiplied with a Hann taper from 1 to 40 Hz with 1 Hz step, from which we computed the PSI at each inter-regional channel pair within the same hemisphere in each participant (i.e., one channel from the anterior/posterior hippocampus and the other from the amygdala) and pooled all possible channel pairs between the hippocampus and the amygdala for each participant. To correct for any spurious results, we randomly shuffled the trials and recomputed the PSI at each channel pair. This step was repeated 200 times to create normal distributions of channel pair-resolved null PSI data. To construct a directional effect of the amygdala-hippocampus on a population level, we averaged the raw PSI across channel pairs and participants. Correspondingly, the null distributions were also averaged across channel pairs and participants. Consequently, the raw PSI outputs can be compared to the distribution of null PSI to derive a *z*-score in the frequency band 1–40 Hz (for a similar approach, see Solomon et al.[28]). Significant PSI was thresholded at $|z| > 1.96$, in which the hippocampus leads were defined as $z > 1.96$ and the amygdala leads as $z < -1.96$, as previous studies performed[12,46].

### Decoding analysis

To investigate whether the neural data in the amygdala and the hippocampus was modulated by WM load, we computed the EED, EMS, and PSI from trials of load 4, 6 and 8, separately. Here we used a linear support vector machine (SVM)[29] as a classifier to decode the WM load (load 4, 6 and 8). SVM is widely used in decoding analyses in neuroimaging studies[48] because of its suitability for analyses with a relatively small number of samples. The SVM analyses in the current study are conducted by LIBSVM package[29] in MATLAB. The details of our decoding analyses were as follows:

(A) EED pattern: We first considered the EED patterns within the amygdala and the hippocampus to decode the WM load. For each load, the EED pattern at trial-pairs level for all participants were merged as the data (100 trial-pairs × 14 participants = 1400 samples) used in the classification. The EED pattern included 201 × 201 = 40401 (201 denotes time windows during encoding) values, and these values were converted to a feature vector. Then, we split 70% of the data from each load and merged them across all loads as the *training* data set. The remaining data were pooled across all loads as the *testing* data set. Meanwhile, to reduce the feature dimensionality, principal component analysis (*PCA*) was applied to the *training* data set to keep several

principal components (*K* components) that explained 99% of the variance in the data. We also transformed the *testing* data set with the *PCA* matrix that was already fitted to the *training* data set. In total, we had the *training* data set with (980 × 3) samples × *K* features and the *testing* data set with (420 × 3) samples × *K* features. We trained an SVM classifier with a linear kernel with a cost equal to one. This procedure was replicated by 100 times for the cross-validation. Accuracy of the classifier as performance measures was averaged across 100 cross-validations. We performed this classification by using EED features in the amygdala and the hippocampus, separately.

(B) EMS pattern: Next, we decoded the WM load by using the EMS features within the amygdala and the hippocampus, respectively. Decoding analyses were performed as described for the EED patterns. The EMS patterns included 201 × 301 = 60501 values and were converted to a feature vector. Similar as described for EED patterns, for each load, we combined the EMS features at trial level for all participants as the data, then split 70% of the data as the *training* data set (*N* samples), then applied *PCA* to the *training* data set to *K* components that explained 99% of the variance, then fed the features (($N_{load4} + N_{load6} + N_{load8}$) samples × *K* features) into a linear SVM classifier for *training*, and tested the model on the remaining pooled data set that already applied *K* components matrix to the *testing* data set (($M_{load4} + M_{load6} + M_{load8}$) samples × *K* features). The accuracy was averaged across 100 cross-validations. Again, this classification was performed by using EMS features in the amygdala and the hippocampus, separately.

(C) PSI pattern: Then we tested whether WM load could be decoded by the directional connectivity from the hippocampus leads as well as the amygdala leads during encoding and maintenance, separately. We first extracted the directional connectivity from the hippocampus leads as well as the amygdala leads based on the *z*-scored PSI at each channel-pair for each participant. Then, the directional connectivity was vectorized (*K* features) at channel-pairs level for all participants and was merged as the data for each load classification (172 channel-pairs). Again, we split the directional connectivity for each load using 70/30 split, with 70% of the data (120 samples) for *training* and the remaining data (52 samples) for *testing*. Then, we fed the *training* data set ((120 × 3) samples × *K* features) into the SVM classifier and tested it on the *testing* data set ((52 × 3) samples × *K* features). This procedure was replicated 100 times for cross-validation. In total, we performed four classifications using PSI features from the hippocampus leads and from the reverse direction during encoding and maintenance, separately.

For each decoding analysis, we used a nonparametric permutation *test* to evaluate the significance of the decoding accuracy difference between two regions or directions. Specifically, we shuffled the labels 200 times, and in each shuffling, we calculated decoding accuracy differences between two regions or directions, resulting a null distribution that encompasses data. *P* values were computed by comparing observed decoding accuracy difference with the entire distribution of null difference in decoding accuracy. We considered $P < 0.05$ to be significant.

### Reporting summary

Further information on research design is available in the Nature Portfolio Reporting Summary linked to this article.

## Data availability

The raw data used in this study[49] have been downloaded from a public database under accession link https://doi.gin.g-node.org/10.12751/g-node.d76994/. The task is freely available for download at http://www.

neurobs.com/ex_files/expt_view?id=266. Links to updates and further data sets can be found at https://hfozuri.ch. Source data are provided with this paper.

## Code availability

FieldTrip and EEGLAB toolbox was used for processing the iEEG data. The custom codes used for this study are available at https://doi.org/10.5281/zenodo.7804834[50].

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

## Acknowledgements

This work received support from the following sources: STI2030-Major Projects (Grant No. 2021ZD0200200 to T.J.), National Natural Science Foundation of China (grant Nos. 32271085 to J.L., 82151307 to T.J.), Science Frontier Program of the Chinese Academy of Sciences (grant No. XDBS01030200 to T.J.), Open Research Fund of the State Key Laboratory of Cognitive Neuroscience and Learning (CNLYB2004 to J.L.), the Swiss National Science Foundation (funded by SNSF 204651 to J.S.). This work was partially supported by the Key Research Project of Zhejiang Lab (No. 2022KI0AC02 to T.J., No. 2022ND0AN01 to T.J.). The authors thank Rhoda E. Perozzi and Edmund F. Perozzi, PhDs, for English and content editing assistance. The authors also thank Vasileios Dimakopoulos for data collection.

## Author contributions

Conceptualization, J.L., D.C., T.J. and J.S.; methodology, J.L. and D.C.; data collection, J.S.; patient care, L.I., and L.S.; formal analysis: J.L., D.C., and X.X.; writing—original draft, J.L. and D.C.; writing—review and editing, T.J., J.S., and S.Y.; funding acquisition, T.J., J.S., and J.L.; resources, T.J., L.I., and J.S.; supervision, T.J. and J.S.

## Competing interests

The authors declare no competing interests.
