## [Peer Review File · Nature Communications]

Functional Specialization and Interaction in the Amygdala-Hippocampus Circuit during Working Memory ProcessingReviewers' comments:

Reviewer #1 (Remarks to the Author):

The investigators report the results of amygdala and hippocampal recordings in 14 human intracranial EEG subjects performing a working memory paradigm based on recognition of letters following a 3 second delay. The analyses focus on differences between the amygdala and hippocampus for correctly recognized items. The overall context of this research is that the contributions of MTL structures to working memory processing is controversial, and therefore identifying new insights based on electrophysiological recordings is worthwhile.

Regarding the paradigm – it is a reasonable task for working memory, although it is easier than alternatives such as N—back. Subject performance on the task is near ceiling (92% correct). There is no analytical approach based on working memory load apart from success. This leads to my principal criticism of this work, which is that the analyses focus exclusively on correct trials rather than a comparison of successful versus unsuccessful memory events. There is no analysis related to the number of items in the stimulus, although this parameter was manipulated in the paradigm. The authors are aware of this issue, and they include analyses that seek to establish a functional effect of the patterns they observe. However, their approach shows only that when they restrict analysis to incorrect trials, they do not find a significant difference. However, this is not the same as demonstrating a functional association for the oscillatory patterns that they observe. More specifically, for the power difference between hippocampus and amygdala for encoding and maintenance, the overall pattern for incorrect trials is identical as for correct trials, but it doesn't reach significance, possibly related to the small number of incorrect trials. The downsampling procedure does not solve this problem, although it does show that the observed patterns for correct trials may be robust even if smaller trial numbers were included. The lack of a functional difference for oscillatory patterns means that inter—regional differences in visual processing may confound the encoding/maintenance differences for example. More generally, it means that hippocampal activity, for example, may be related to processes unrelated to working memory.

The conclusion that less similarity across encoding items in the amygdala is related to representational specificity is somewhat overstated. There is no analysis based on more similar versus dissimilar items to establish behavioral conclusions from the observation of encoding/encoding similarity patterns.

Also, I don't entirely understand the decision to focus only on 1-40 Hz frequencies for such an analysis, since there is activity in higher gamma ranges which has been linked to reinstatement patterns in the hippocampus.

The encoding/maintenance similarity analysis shows greater similarity in the hippocampus as compared to amygdala. I don't understand why encoding/retrieval similarity is not the focus of a reinstatement analysis, as this is the most analogous (presence of visual stimulus) in the behavioral paradigm.

Reviewer #2 (Remarks to the Author):

In this study, Li et al. used intracranial EEG in epilepsy patients to determine the role of the hippocampus and the amygdala as well as the coordination between these areas in working memory processes. The authors report that amygdala representations were distinct and decreased from encoding to maintenance, whereas hippocampal representations remained stable during the maintenance phase. The authors further suggest an information flow from the hippocampus to the amygdala.

Overall, this is an interesting and timely topic. The used intracranial EEG recordings in humans are methodologically challenging and provide a powerful tool to enhance our understanding of brain-behavior relationships in humans, with excellent temporal and spatial resolution.

However, I have identified several issues that dampened my enthusiasm. My main concerns relate to whether the conclusions of the authors are sufficiently supported by the data.

1) The rationale of the study and its additional value beyond the already existing studies on this topic were not sufficiently clear to me. As the authors indicate in their introduction, there are already several intracranial EEG studies on the role of the hippocampus and amygdala in working memory. How exactly does the present study go beyond these previous reports? The authors write "the more general significance of amygdala and hippocampal activity for dealing with WM information and their specific contribution to WM is still unknown" and later "it remains to be established whether the engagement of the two structures is coordinated or independent". This is rather vague. What is the specific aim and contribution of this study beyond the already existing studies? Also, I was missing specific hypotheses.

2) The Sternberg paradigm included set sizes of 4, 6 or 8 stimuli. From previous research on working memory (including studies using the Sternberg paradigm), we know that set size has a significant impact on working memory performance. I was therefore somewhat surprised to see that the factor set size was not included in the present analyses. I would like to see the effect of set size on behavioural performance and ideally also whether the neural data are influenced by the factor set size.

3) I am not entirely convinced that the conclusions of the authors are sufficiently backed-up by the data.

(a) On page 8, the authors report first that power in the hippocampus was during maintenance was higher than during encoding at 3-13 Hz. Here, it was not entirely clear to me how the authors arrived at this frequency range and whether proper corrections for multiple testing (across the frequency range of 1-40Hz) was performed. More importantly, after the authors identified the significant effect in the 3-13Hz range, they then continue to separately analyse the brain region by task phase interaction specifically for this frequency range. It seemed to me that this strategy qualifies as 'double dipping' and an analysis that explicitly takes the factor frequency range into account would seem more appropriate in my view. Please clarify.

(b) On the same page, the authors conclude that "the amygdala was preferentially engaged in encoding". This is at least misleading because there was no power differences between encoding and maintenance in the amygdala ($p=.20$).

(c) On page 9 and 10, the authors report that the encoding-encoding dissimilarity (EED) was higher for the amygdala than for the hippocampus and they conclude that "the amygdala represented distinct WM information in a more specific neural pattern during encoding". However, I do not think that this conclusion is necessarily warranted. The authors assume that the increased EED reflects stimulus-specificity. However, it might also reflect increased process dynamics (i.e. that the contribution of the amygdala is generally more variable). In the present set-up, it remains difficult to really show that this dissimilarity is due to stimulus-specificity. Perhaps, the authors could leverage the different set sizes here and test whether the EED in the amygdala is modulated by set-size.

(d) On page 11, the authors analyze the correlation between the difference of EED in amygdala and hippocampus and the difference of EMS in hippocampus and amygdala and conclude from a significant correlation that "in subjects in which the amygdala represented more WM information, the hippocampus maintained the representation more stably". In my view, the correlation of these differences scores is difficult to interpret (e.g. a high EED in the amygdala may directly lead to a reduced EMS, which might drive the observed correlation) and the authors' conclusion therefore not necessarily warranted.

(e) On page 14, the authors suggest that the specific pattern in the amygdala and hippocampus (as well as their coordination) would be related to successful working memory performance. This conclusion is based on the findings that the results obtained for correct trials were not observed for incorrect trials. As the authors note themselves, this differences between correct and incorrect trials might be due to the unequal number of correct (>90 percent of all trials) and incorrect trials. The authors offer bootstrapping analyses as control analysis, which is helpful and appreciated. However, while these control, analyses show that the reported findings can be observed in a relatively low number of correct trials, they do not rule out that very similar findings might be present for incorrect trials if there were

more of these trials (in fact, even for the relatively few incorrect trials, there are trends). Thus, the conclusion of the authors should be toned down in the sense that the effects appeared to be stronger for correct than for incorrect trials. Moreover, I thought that a more direct way to link neural activity patterns and behavioural performance would be via direct activity-performance correlations at the level of trials or groups of trials. Furthermore, an inclusion of the different set sizes (which should result in different levels of performance) might be helpful here.

In sum, the authors report several complex analyses and parameters, the interpretation of which is in my view at least less straightforward than suggested by the authors and would need further clarifications.

4) Finally, the authors suggest on the one hand that the amygdala was preferentially involved at encoding (but see my comment above) and the hippocampus during maintenance and that there would be unidirectional information flow from the hippocampus to the amygdala during encoding and maintenance. I found these findings difficult to reconcile. If the amygdala contain stimulus-specific representations and is preferentially engaged during encoding, how should then the hippocampus inform the amygdala (and not vice versa). I think these findings would need a bit more of discussion.

Reviewer comments

We thank all expert Reviewers for their encouraging and constructive comments, which have undoubtedly improved the quality of our manuscript. Here, we provide a point-by-point response to all comments. Our responses are in blue, with text that was in the original submission in *blue italics*, and new text in red.

1 Reviewer #1:

The investigators report the results of amygdala and hippocampal recordings in 14 human intracranial EEG subjects performing a working memory paradigm based on recognition of letters following a 3 second delay. The analyses focus on differences between the amygdala and hippocampus for correctly recognized items. The overall context of this research is that the contributions of MTL structures to working memory processing is controversial, and therefore identifying new insights based on electrophysiological recordings is worthwhile.

Reply: We thank the reviewer for this appreciation.

1.1 Paradigm: Number of items in the stimulus

Regarding the paradigm – it is a reasonable task for working memory, although it is easier than alternatives such as N—back. Subject performance on the task is near ceiling (92% correct). There is no analytical approach based on working memory load apart from success. This leads to my principal criticism of this work, which is that the analyses focus exclusively on correct trials rather than a comparison of successful versus unsuccessful memory events. There is no analysis related to the number of items in the stimulus, although this parameter was manipulated in the paradigm.

Reply: We have decided for this working memory (WM) paradigm because it separates encoding from maintenance, which we analyze separately in our study.

Action: Following both Reviewer’s suggestion, we have now analyzed the effect of number of items (workload) in the study. The new analysis pipeline is outlined in the new Panel D in Fig. 1.

Fig. 1 (D) Decoding analysis: we used machine learning analysis to investigate whether WM load could be decoded by encoding-encoding dissimilarity (EED), encoding-maintenance similarity (EMS), or Granger Causality (GC) features. (Figure legends, page 33)

We describe these new findings in the Results at pages 12-13:

Functional specialization and interaction within the amygdala-hippocampal circuit predict WM load

...Then, to address whether the neural representation and connectivity patterns within the amygdala-hippocampal circuit could predict WM load, we made decoding analyses with the patterns of EED, EMS and GC from trials of load 4, 6 and 8, separately. We used a linear support vector machine (SVM)² classifier to predict WM load and tested with leave-one-out cross-validation by using the EED, EMS, or GC features, respectively. The statistical significance of the classification accuracy was determined by comparing the original accuracy with a null distribution created by using a randomized classifier by permuting the labels 1000 times (see details in **Methods**).

We first investigated whether WM load could be decoded by the EED or EMS features in the amygdala or the hippocampus, respectively. We extracted the representative patterns with 3 loads from all the participants as features and fed them into the SVM classifier to classify WM load. Significant EED-based decoding results were found in the amygdala but not in the hippocampus (permutation test against scrambled data, $p < 0.05$, see **Fig. 5(A)**). Decoding accuracy was significantly above chance level by using the EMS features in the hippocampus but not in the amygdala (permutation test against scrambled data, $p < 0.05$, **Fig. 5(B)**).

We followed up by investigating whether the inter-regional interaction between the amygdala and the hippocampus could predict WM load. GC values at 1-40 Hz from

both directions (from the amygdala to the hippocampus, and vice versa) in the encoding and maintenance periods were extracted as features to decode the WM load. During the encoding period, WM load could be decoded by using GC features from the amygdala to the hippocampus (permutation test against scrambled data, $p = 0.05$) but not by that from the opposite direction as predictors ($p > 0.05$; **Fig. 5(C)** left). During maintenance, WM load could be decoded by the GC features from the hippocampus to the amygdala (permutation test against scrambled data, $p < 0.05$) but not by that from the opposite direction ($p > 0.05$; **Fig. 5(C)** right).

Taken together, our collective results that WM load can be predicted by representational features in the amygdala during encoding and in the hippocampus during maintenance, and by information flow from the amygdala during encoding and that from the hippocampus during maintenance indicated that the amygdala contributed to WM encoding and the hippocampus participated in WM maintenance in a load-sensitive manner.

Fig. 5 Decoding accuracy of WM load using SVM classifier. (A) Decoding accuracy using the EED patterns within the amygdala (red) and the hippocampus (blue). Only using the features from the amygdala rather than the hippocampus was able to decode the WM load. * $p < 0.05$. (B) Decoding accuracy using the EMS patterns within the amygdala (red) and the hippocampus (blue). Only using the features from the hippocampus rather than the amygdala could decode the WM load. * $p < 0.05$. (C)

Decoding accuracy using the GC features from both directions during encoding (left) and maintenance (right). WM load could be decoded with the GC features from the amygdala to the hippocampus during encoding (left) and the GC features from the hippocampus to the amygdala during maintenance (right). (Figure legends, page 38)

We describe the new decoding analysis in the Methods at pages 25-27:

To investigate whether the neural data in the amygdala and the hippocampus was modulated by WM load, we computed the EED, EMS, and GC values from trials of load 4, 6 and 8, separately. Here we used a linear support vector machine (SVM) ² as a classifier to decode the WM load (load 4, 6 and 8). SVM is widely used in decoding analyses in neuroimaging studies ³ because of its suitability for analyses with a relatively small number of samples. The SVM analyses in the current study are conducted by the COSMOMVPA package ⁴ in MATLAB. The details of our decoding analyses were as follows:

(A) EED: We first considered the EED patterns within the amygdala and the hippocampus to decode the WM load. The EED patterns were extracted from trials with load 4, 6 and 8, separately. For each participant and each load, the EED pattern included $201 \times 201 = 40401$ (201 denotes time windows during encoding) values, and these values were converted to a feature vector. We used principal component analysis (PCA) to reduce the number of EED features to M principal components that explained 99% of the variance of the data. To increase the impact of the analysis on a larger population, we merged the data from all the participants ($N = 14$) and all the loads to perform the classification across the participants. We trained an SVM classifier with a linear kernel with a cost equal to one. Then we used the leave-one-out cross-validation method to ensure accurate estimates of predictive validity. Specifically, the feature vectors from $N-1$ participants were used as a training dataset ($(N-1) \times M \times 3$ loads) and we tested the SVM classifier on the remaining dataset from the left participant. Then, we repeat the train-test method N times such that each time one of the N participants is tested and the rest $N-1$ participants are used together as a training set. The accuracy of the classifier was averaged across all cross-validations as a measure of performance. We performed this classification by using EED features in the amygdala and the hippocampus, separately.

(B) EMS: Next, we decoded the WM load by using the EMS features in the amygdala and the hippocampus, respectively. For each participant at each load, the

EMS values ($201 \times 301 = 60501$, 201 denotes time windows during encoding and 301 means time windows during maintenance) were converted to a feature vector, and the number of the features was reduced using PCA to M principal components. The decoding and cross-validation methods were similar to that mentioned in EED decoding, except that the input features are replaced with EMS features. Again, this classification was performed by using EMS features in the amygdala and the hippocampus, separately.

(C) GC: Then we tested whether WM load could be decoded by GC features from the hippocampus to the amygdala and those from the reverse direction during the encoding and the maintenance period, separately. For each participant and each direction, the GC features were extracted at the frequency band 1-40 Hz from trials with load 4, 6 and 8, separately. Again, the number of these features was reduced using PCA and fed into a linear SVM classifier and tested with leave-one-out cross-validation. We performed this classification process by using GC features from the hippocampus to the amygdala and from the reverse direction, during encoding and maintenance, separately.

For each decoding analysis, we used a nonparametric permutation approach to test the significance of the accuracy values. Specifically, we permuted the load labels 1000 times and then calculated the corresponding decoding accuracy to create a null distribution. For each kind of features (EED or EMS or GC), we took the larger value of the null distribution across both structures (for GC features, those are both directions) as a final null-distribution to correct for multiple comparisons across structures, as a previous study did³. The original decoding accuracy value (found from a classifier with true labels) was considered significant when the observed accuracy exceeded the 95th percentile of the null distribution ($p < 0.05$).

1.2 Analysis of incorrect trials

The authors are aware of this issue, and they include analyses that seek to establish a functional effect of the patterns they observe. However, their approach shows only that when they restrict analysis to incorrect trials, they do not find a significant difference. However, this is not the same as demonstrating a functional association for the oscillatory patterns that they observe. More specifically, for the power difference

between hippocampus and amygdala for encoding and maintenance, the overall pattern for incorrect trials is identical as for correct trials, but it doesn't reach significance, possibly related to the small number of incorrect trials. The downsampling procedure does not solve this problem, although it does show that the observed patterns for correct trials may be robust even if smaller trial numbers were included.

Reply: We agree that the small number of incorrect trials may lead to the lack of significant difference.

Action: Following the Reviewer's comment, we now removed the analyses on incorrect trials from the manuscript. Instead, we added analyses on WM load where the number of trials with different loads is balanced.

1.3 Inter-regional differences in visual processing

The lack of a functional difference for oscillatory patterns means that inter-regional differences in visual processing may confound the encoding/maintenance differences for example. More generally, it means that hippocampal activity, for example, may be related to processes unrelated to working memory.

Reply: Thanks for this comment. We agree that we should exclude the possibility that our results are due to differences in processes unrelated to WM, such as visual processing.

Action: To test this possibility, we computed the event-related potentials (ERPs) induced by the visual stimuli for the amygdala and the hippocampus during WM encoding period, in the amygdala and the hippocampus for each participant, respectively. No difference in ERP was found between the two regions. These findings indicated that our results could not be explained by differences in visual processing. This result was added to a **new supplementary figure Fig. S2**. We now write in the Results at pages 9-10:

Besides, to further exclude the possibility that the inter-regional differences we found were due to visual processing rather than WM, we computed event-related potentials (ERP) induced by the visual stimuli in the encoding period from these two regions, respectively. Results showed no difference between the averaged amplitude of ERP in the amygdala and that in the hippocampus (paired t -test, $p = 0.22$, $t(13) = -1.29$, **Fig. S2(C)**). Besides, the ERP showed no significant difference at each time point throughout the encoding period (cluster-based permutation test, $p > 0.05$, **Fig. S2(D)**).

New supplementary figure Fig. S2 (C) Averaged amplitude of event-related potential (ERP) for the hippocampus (blue) and the amygdala (red) in encoding. No difference was found between the two regions. (D) The ERP at each time point for the hippocampus (blue) and the amygdala (red) throughout the encoding period. No difference was found at each time point between the two regions. (Figure legends, page 40)

1.4 Representational specificity

The conclusion that less similarity across encoding items in the amygdala is related to representational specificity is somewhat overstated. There is no analysis based on more similar versus dissimilar items to establish behavioral conclusions from the observation of encoding/encoding similarity patterns.

Reply: We agree that the lack of the same items is a limitation of our study and hinders the comparison between the same and different item pairs.

We are now more cautious to interpret the smaller similarity across encoding trials in the amygdala.

Action: Following the Reviewer's comment, we have now removed the words "representational specificity" from the manuscript. We now write in the Results at page 9:

As the letter strings were different across trials, these findings indicated that the activity patterns for different items had a larger distance among each other within the amygdala, whereas the hippocampus showed overlapping representations across different items with reduced neural dissimilarity.

1.5 Focus on frequencies 1-40 Hz for analysis

Also, I don't entirely understand the decision to focus only on 1-40 Hz frequencies for such an analysis, since there is activity in higher gamma ranges which has been linked to reinstatement patterns in the hippocampus.

Reply: We agree that the frequency band used in the representational analysis is important.

Action: Following the Reviewer' comment, we have now added further analyses that we present in a **new supplementary figure Fig. S1**. We now write in the Results at page 8:

Neural activity on 1-40 Hz frequencies was included in representational analyses. We chose this frequency range for the following two reasons. First, both the amygdala and the hippocampus showed elevated activity in the low-frequency range 1-40 Hz (**Fig. S1(A)**). Second, the averaged z-scored power on 1-40 Hz was significantly above zero at most time points during WM processing, while only a few time points showed significantly elevated activity for the averaged z-scored power on 40-100 Hz (**Fig. S1(B)**). These findings are in line with our previous study ⁶.

New supplementary figure Fig. S1 (A) The time-frequency plot of z-scored power in the hippocampus (left) and the amygdala (right). Warmer color denotes higher z-scored power. (B) Averaged z-scored power across 1-40 Hz (left) and 40-100 Hz (right) at each time point in the hippocampus (blue) and the amygdala (red). Blue and red lines at the top denote the time point with an activity significantly above zero (cluster-based permutation test, $p < 0.05$). (Figure legends, page 39)

1.6 Reinstatement analysis

The encoding/maintenance similarity analysis shows greater similarity in the hippocampus as compared to amygdala. I don't understand why encoding/retrieval similarity is not the focus of a reinstatement analysis, as this is the most analogous (presence of visual stimulus) in the behavioral paradigm.

Reply: We agree that in some studies of episodic memory, the reinstatement refers specifically to recognition-related reactivation, thus focusing on the retrieval period ⁷, ⁸, we here are neutral about this period. WM describes our capacity to actively maintain information for a short period of time. For the past decades, studies tried to investigate how information is maintained in working memory in the absence of external stimuli, namely, the neural activity during the maintenance period ^{9, 10}. Our study also focused on how the amygdala and the hippocampus maintain the representation of the external stimulus in WM. Therefore, we studied the encoding-maintenance representational similarity, as a previous study of working memory did ¹¹. We agree with the reviewer that “reinstatement” will introduce unnecessary confusion.

Action: Following the Reviewers' comment, we now refrain from using the word “reinstatement” and use “**encoding-maintenance similarity (EMS)**” instead.

2 Reviewer #2:

In this study, Li et al. used intracranial EEG in epilepsy patients to determine the role of the hippocampus and the amygdala as well as the coordination between these areas in working memory processes. The authors report that amygdala representations were distinct and decreased from encoding to maintenance, whereas hippocampal representations remained stable during the maintenance phase. The authors further suggest an information flow from the hippocampus to the amygdala. Overall, this is an interesting and timely topic. The used intracranial EEG recordings in humans are methodologically challenging and provide a powerful tool to enhance our understanding of brain-behavior relationships in humans, with excellent temporal and spatial resolution. However, I have identified several issues that dampened my enthusiasm. My main concerns relate to whether the conclusions of the authors are sufficiently supported by the data.

Reply: We thank the reviewer for this appreciation.

2.1 Rationale of the study

The rationale of the study and its additional value beyond the already existing studies on this topic were not sufficiently clear to me. As the authors indicate in their introduction, there are already several intracranial EEG studies on the role of the hippocampus and amygdala in working memory. How exactly does the present study go beyond these previous reports? The authors write “the more general significance of amygdala and hippocampal activity for dealing with WM information and their specific contribution to WM is still unknown” and later “it remains to be established whether the engagement of the two structures is coordinated or independent”. This is rather vague. What is the specific aim and contribution of this study beyond the already existing studies? Also, I was missing specific hypotheses.

Reply: We thank the Reviewer for encouraging us to formulate our rationale more clearly. While previous studies described a general involvement of the amygdala and the hippocampus in working memory (WM) processes by using univariate analysis, the functional specialization and the inter-regional communication remained under-explored.

Regarding functional specialization, the specific role of amygdala and hippocampus during different WM periods remained unclear.

Regarding the inter-regional communication, previous reports on the amygdala-hippocampal interaction have focused on emotional memory¹². Whether and how the two structures interact during WM processing of non-emotional contents remained unclear. Therefore, we used multivariate representational and classification analyses on intracranial recordings to address these questions.

Action: Following the Reviewer' suggestion, we have now thoroughly re-written the Introduction Section.

Various tasks in our everyday lives require working memory (WM), for example, temporarily remembering a telephone number that you were just told, or visualizing and holding numbers and symbols in mind to solve a math problem. WM refers to a cognitive system storing information in an active and readily available state for a short period¹³. In humans, several brain areas are thought to be essential for WM¹⁴. Here we focus on two areas: the amygdala and the hippocampus. The amygdala is classically associated with emotional processing¹⁵, while recent studies also showed that the amygdala has multidimensional response properties¹⁶ and plays a role even in memorizing non-emotional stimulus material¹⁷. The hippocampus is typically studied for its role in long-term memories¹⁸. However, converging evidence points that the amygdala and the hippocampus are involved in WM¹⁹, such as persistent neural firing^{1, 20, 21} and elevated hippocampal activation^{6, 22} during WM processing. These studies suggested a general involvement of these areas in WM. However, their specific role in different WM phases has not been established.

Rather than looking at the mean level of activity, multivariate representational analysis methods allow the detection of specific patterns of activity and may be more informative about the representation of specific stimulus (see ref.²³ for a review). Previous studies identified two properties related to memory performance, the representational dissimilarity among different stimulus²⁴ and the representational stability between different memory periods¹¹. The amygdala is known as a detector of goal-related stimuli²⁵ and receives major projections from the anterior temporal lobe²⁶ that convey highly processed object information²⁷. On the other hand, the hippocampus is crucial for memory consolidation²⁸. For instance, recent studies suggested notable overlap in representational patterns between the encoding and the post-encoding period in the hippocampus^{11, 29}. Therefore, we hypothesized a functional specialization in the amygdala and the hippocampus in WM encoding and

maintenance periods. However, no study has yet simultaneously tracked amygdala and hippocampal representations in humans. Whether the perceptual representations differed in the amygdala and the hippocampus, whether they changed from encoding to maintenance phase remained unclear.

Many current theories view WM relies on functionally interconnected brain areas ¹⁴. This raises a question: how do the amygdala and the hippocampus work together to support WM? The inter-regional communications have started to be addressed. At the anatomical level, tract tracing studies have uncovered structural connections between the amygdala and the hippocampus ³⁰. At the functional level, electrical stimulation of the hippocampus can induce synaptic plasticity in the amygdala in rodent studies ³¹, later human studies indicated that stimulation of the amygdala led to increased power in the hippocampus ¹⁷. The structural and electrophysiological evidence suggests inter-regional information communication. However, studies of inter-regional communications between the amygdala and hippocampus primarily focused on emotional memory ^{29, 32}, not WM processing with non-emotional contents. Here we hypothesized that the two areas interact and transfer WM related information during WM.

To address these issues, we recorded iEEG simultaneously from the amygdala and the hippocampus in human epilepsy patients while they performed a WM task. By combining the high temporal resolution of human iEEG recordings with a variety of approaches including representational similarity analysis (**Fig.1(A-B)**), information flow analysis (**Fig.1(C)**) and neural pattern classification analysis (**Fig.1(D)**), the current study examined the aspects of memory representations in the amygdala and the hippocampus and their interactions that contribute to WM. We found that the amygdala forms distinct mnemonic representations during encoding while the hippocampus keeps stable representations from encoding to maintenance. Next, we observed enhanced inter-regional information transfer during both encoding and maintenance. Finally, the functional specialization and interaction patterns were predictive of WM load.

2.2 Set size analysis

The Sternberg paradigm included set sizes of 4, 6 or 8 stimuli. From previous research on working memory (including studies using the Sternberg paradigm), we know that set size has a significant impact on working memory performance. I was therefore somewhat surprised to see that the factor set size was not included in the present analyses. I would like to see the effect of set size on behavioural performance and ideally also whether the neural data are influenced by the factor set size.

Reply: We appreciated for this valuable suggestion. We agree that set size has a significant impact on working memory performance. Therefore, we made new analysis on WM accuracy with different WM load as well as on the prediction of WM load using neural features we found.

Action: Following both Reviewer's suggestion, we have now analyzed the effect of number of items (workload) in the study. The new analysis pipeline is outlined in the new Panel D in Fig. 1.

Fig. 1 (D) Decoding analysis: we used machine learning analysis to investigate whether WM load could be decoded by encoding-encoding dissimilarity (EED), encoding-maintenance similarity (EMS), or Granger Causality (GC) features. (Figure legends, page 33)

We describe these new findings in the Results at pages 12-13:

Functional specialization and interaction within the amygdala-hippocampal circuit predict WM load

In this study, we first tested the effect of set size (WM load) on the participants' response accuracy. We found that the accuracy of WM decreased from load 4 (mean \pm S.D.: 98.04% \pm 1.91%) to load 6 (90.78% \pm 5.36%) and 8 (85.36% \pm 5.89%) (repeated-measures analysis of variance (ANOVA), $F(2,26) = 42.71$, $p < 0.001$, **Fig. 2(B)**). This finding indicates that the behavioral performance was modulated by WM

load, which is in line with previous study that the factor of load had a significant impact on working memory performance¹.

Fig. 2 (B) Accuracy of load 4, 6 and 8 across all participants. Each dot denotes each participant and each dotted line connects an individual. ** $p < 0.01$ (Figure legends, page 34)

Then, to address whether the neural representation and connectivity patterns within the amygdala-hippocampal circuit could predict WM load, we made decoding analyses with the patterns of EED, EMS and GC from trials of load 4, 6 and 8, separately. We used a linear support vector machine (SVM)² classifier to predict WM load and tested with leave-one-out cross-validation by using the EED, EMS, or GC features, respectively. The statistical significance of the classification accuracy was determined by comparing the original accuracy with a null distribution created by using a randomized classifier by permuting the labels 1000 times (see details in **Methods**).

We first investigated whether WM load could be decoded by the EED or EMS features in the amygdala or the hippocampus, respectively. We extracted the representative patterns with 3 loads from all the participants as features and fed them into the SVM classifier to classify WM load. Significant EED-based decoding results were found in the amygdala but not in the hippocampus (permutation test against scrambled data, $p < 0.05$, see **Fig. 5(A)**). Decoding accuracy was significantly above chance level by using the EMS features in the hippocampus but not in the amygdala (permutation test against scrambled data, $p < 0.05$, **Fig. 5(B)**).

We followed up by investigating whether the inter-regional interaction between the amygdala and the hippocampus could predict WM load. GC values at 1-40 Hz from both directions (from the amygdala to the hippocampus, and vice versa) in the

encoding and maintenance periods were extracted as features to decode the WM load. During the encoding period, WM load could be decoded by using GC features from the amygdala to the hippocampus (permutation test against scrambled data, $p = 0.05$) but not by that from the opposite direction as predictors ($p > 0.05$; **Fig. 5(C)** left). During maintenance, WM load could be decoded by the GC features from the hippocampus to the amygdala (permutation test against scrambled data, $p < 0.05$) but not by that from the opposite direction ($p > 0.05$; **Fig. 5(C)** right).

Taken together, our collective results that WM load can be predicted by representational features in the amygdala during encoding and in the hippocampus during maintenance, and by information flow from the amygdala during encoding and that from the hippocampus during maintenance indicated that the amygdala contributed to WM encoding and the hippocampus participated in WM maintenance in a load-sensitive manner.

Fig. 5 Decoding accuracy of WM load using SVM classifier. (A) Decoding accuracy using the EED patterns within the amygdala (red) and the hippocampus (blue). Only using the features from the amygdala rather than the hippocampus was able to decode the WM load. * $p < 0.05$. (B) Decoding accuracy using the EMS patterns within the amygdala (red) and the hippocampus (blue). Only using the features from the hippocampus rather than the amygdala could decode the WM load. * $p < 0.05$. (C) Decoding accuracy using the GC features from both directions during encoding (left)

and maintenance (right). WM load could be decoded with the GC features from the amygdala to the hippocampus during encoding and the GC features from the hippocampus to the amygdala during maintenance. (Figure legends, page 38)

We describe the new decoding analysis in the Methods at pages 25-27:

To investigate whether the neural data in the amygdala and the hippocampus was modulated by WM load, we computed the EED, EMS, and GC values from trials of load 4, 6 and 8, separately. Here we used a linear support vector machine (SVM) ² as a classifier to decode the WM load (load 4, 6 and 8). SVM is widely used in decoding analyses in neuroimaging studies ³ because of its suitability for analyses with a relatively small number of samples. The SVM analyses in the current study are conducted by the COSMOMVPA package ⁴ in MATLAB. The details of our decoding analyses were as follows:

(D) EED: We first considered the EED patterns within the amygdala and the hippocampus to decode the WM load. The EED patterns were extracted from trials with load 4, 6 and 8, separately. For each participant and each load, the EED pattern included $201 \times 201 = 40401$ (201 denotes time windows during encoding) values, and these values were converted to a feature vector. We used principal component analysis (PCA) to reduce the number of EED features to M principal components that explained 99% of the variance of the data. To increase the impact of the analysis on a larger population, we merged the data from all the participants ($N = 14$) and all the loads to perform the classification across the participants. We trained an SVM classifier with a linear kernel with a cost equal to one. Then we used the leave-one-out cross-validation method to ensure accurate estimates of predictive validity. Specifically, the feature vectors from $N-1$ participants were used as a training dataset ($(N-1) \times M \times 3$ loads) and we tested the SVM classifier on the remaining dataset from the left participant. Then, we repeat the train-test method N times such that each time one of the N participants is tested and the rest $N-1$ participants are used together as a training set. The accuracy of the classifier was averaged across all cross-validations as a measure of performance. We performed this classification by using EED features in the amygdala and the hippocampus, separately.

(E) EMS: Next, we decoded the WM load by using the EMS features in the amygdala and the hippocampus, respectively. For each participant at each load, the EMS values ($201 \times 301 = 60501$, 201 denotes time windows during encoding and 301

means time windows during maintenance) were converted to a feature vector, and the number of the features was reduced using PCA to M principal components. The decoding and cross-validation methods were similar to that mentioned in EED decoding, except that the input features are replaced with EMS features. Again, this classification was performed by using EMS features in the amygdala and the hippocampus, separately.

(F) GC: Then we tested whether WM load could be decoded by GC features from the hippocampus to the amygdala and those from the reverse direction during the encoding and the maintenance period, separately. For each participant and each direction, the GC features were extracted at the frequency band 1-40 Hz from trials with load 4, 6 and 8, separately. Again, the number of these features was reduced using PCA and fed into a linear SVM classifier and tested with leave-one-out cross-validation. We performed this classification process by using GC features from the hippocampus to the amygdala and from the reverse direction, during encoding and maintenance, separately.

For each decoding analysis, we used a nonparametric permutation approach to test the significance of the accuracy values. Specifically, we permuted the load labels 1000 times and then calculated the corresponding decoding accuracy to create a null distribution. For each kind of features (EED or EMS or GC), we took the larger value of the null distribution across both structures (for GC features, those are both directions) as a final null-distribution to correct for multiple comparisons across structures, as a previous study did³. The original decoding accuracy value (found from a classifier with true labels) was considered significant when the observed accuracy exceeded the 95th percentile of the null distribution ($p < 0.05$).

2.3 Validity of conclusions

I am not entirely convinced that the conclusions of the authors are sufficiently backed-up by the data.

2.3.1 Encoding at 3-13 Hz

On page 8, the authors report first that power in the hippocampus was during maintenance was higher than during encoding at 3-13 Hz. Here, it was not entirely clear to me how the authors arrived at this frequency range and whether proper

corrections for multiple testing (across the frequency range of 1-40Hz) was performed. More importantly, after the authors identified the significant effect in the 3-13Hz range, they then continue to separately analyses the brain region by task phase interaction specifically for this frequency range. It seemed to me that this strategy qualifies as 'double dipping' and an analysis that explicitly takes the factor frequency range into account would seem more appropriate in my view. Please clarify.

Reply: As mentioned in Reply 2.1, we aimed to study the functional specialization by using multivariate representational analyses, rather than to validate a general involvement of the amygdala and hippocampus in WM by using a univariate power analysis. We now realized the power analysis is indeed somewhat unrelated to the main analyses.

Action: We moved the analysis related to time-frequency power to the **new supplementary figure Fig. S1**. Besides, in the revised version, we removed contrast of the power between two regions, as it is unrelated to the main analyses.

2.3.2 Amygdala in encoding

On page 8, (b) On the same page, the authors conclude that “the amygdala was preferentially engaged in encoding”. This is at least misleading because there was no power differences between encoding and maintenance in the amygdala ($p=.20$).

Reply: We regret this misunderstanding. In the original manuscript, “the amygdala was preferentially engaged in encoding”, we intended to indicate the result “the power in the amygdala was higher than in the hippocampus during encoding ($p =0.017$)”. As mentioned in Reply 2.1, the current study focused on multivariate representational analysis rather than univariate power analysis.

Action: The results addressed here are from power analysis, which is now less relevant and no longer reported in the main analyses. We have now removed this result so as not to obscure the focus.

2.3.3 EED was higher for the amygdala

On page 9 and 10, the authors report that the encoding-encoding dissimilarity (EED) was higher for the amygdala than for the hippocampus and they conclude that “the amygdala represented distinct WM information in a more specific neural pattern during encoding”. However, I do not think that this conclusion is necessarily warranted. The

authors assume that the increased EED reflects stimulus-specificity. However, it might also reflect increased process dynamics (i.e. that the contribution of the amygdala is generally more variable). In the present set-up, it remains difficult to really show that this dissimilarity is due to stimulus-specificity. Perhaps, the authors could leverage the different set sizes here and test whether the EED in the amygdala is modulated by set-size.

Reply: We agree that our interpretation of the less similarity across encoding items “representational specificity” is somewhat speculative at the current stage. We also agree there is an alternative explanation: higher activity fluctuation across trials in the amygdala.

Action: To test this explanation, we calculated the variability (standard error) of the averaged power at 1-40 Hz during the encoding period across trials, in the amygdala and the hippocampus for each participant, respectively. And no difference of the variability was found between these two regions. This result was added to a **new supplementary figure Fig. S2**. We now write in the Results at page 9:

As the letter strings were different across trials, these findings indicated that the activity patterns for different items had a larger distance among each other within the amygdala, whereas the hippocampus showed overlapping representations across different items with reduced neural dissimilarity.

We could think of two possible explanations for the higher EED in the amygdala, one is the specific representation of distinct items. Alternatively, it could also be due to higher activity fluctuation across trials in the amygdala. To test this explanation, we calculated the variability (standard error) of the averaged z-scored power at 1-40 Hz during the encoding period across trials, in the amygdala and the hippocampus for each participant, respectively. Then we compared them using a paired *t*-test. Results showed no difference between the variability in the amygdala and that in the hippocampus (paired *t*-test, $p=0.94$, $t(13)=0.076$, **Fig. S2(A)**). Next, we compared the activity variability at each time point in the encoding period between the amygdala and the hippocampus. Again, the activity variability showed no significant difference at each time point throughout the encoding period (cluster-based permutation test, $p>0.05$, **Fig. S2(B)**). These findings indicated that less similarity across encoding items in the amygdala could not be explained by differences in power variability.

New supplementary figure Fig. S2 (A) Averaged variability of the power at 1-40 Hz during the encoding period across trials, in the hippocampus (blue) and the amygdala (red). Dots denote individual subjects. No difference in the variability was found between the two regions. (B) The power variability across 1-40 Hz at each time point of the encoding period, in the hippocampus (blue) and the amygdala (red). No difference was found at each time point between the two regions. (Figure legends, page 40)

2.3.4 EED and the EMS values

On page 11, the authors analyze the correlation between the difference of EED in amygdala and hippocampus and the difference of EMS in hippocampus and amygdala and conclude from a significant correlation that “in subjects in which the amygdala represented more WM information, the hippocampus maintained the representation more stably”. In my view, the correlation of these differences scores is difficult to interpret (e.g. a high EED in the amygdala may directly lead to a reduced EMS, which might drive the observed correlation) and the authors’ conclusion therefore not necessarily warranted.

Reply: We apologize for the lack of clarity in our descriptions. We are now more clearly illustrating how we computed the EED and the EMS values, respectively.

Action: For EED (encoding-encoding dissimilarity), we calculated the similarity between activity patterns of trial pairs during encoding. The values were subtracted from 1 and then averaged across trial pairs (Panel A in **Fig.1**, the blue box for illustration). For EMS (encoding-maintenance similarity), we computed the similarity of the activity patterns between encoding and maintenance in the same trial, and then averaged them across trials (Panel B in **Fig.1**, the red box for illustration). As the EED and the EMS calculated different things, we speculated that a higher EED cannot guarantee a lower EMS. We could think of a possible explanation, a higher activity fluctuation in the amygdala than that in the hippocampus, which leads to high EED

and low EMS in the amygdala and thus a correlation. We found no difference between the variability between the amygdala and the hippocampus (new supplementary figure **Fig. S2(A-B)**, also see Reply 2.3.3), which speaks against the above-mentioned explanation. In our aim to streamline the manuscript, we have removed the correlation analysis between the representational features from the manuscript.

Fig.1 (A-B): An illustration on methods we used to compute the indices of the EED and the EMS within the amygdala and the hippocampus. Blue dotted box denotes the EED index was calculated between any two trials, and red dotted box represents the EMS index was computed across the same trials.

2.3.5 EED and the EMS values

On page 14, the authors suggest that the specific pattern in the amygdala and hippocampus (as well as their coordination) would be related to successful working memory performance. This conclusion is based on the findings that the results obtained for correct trials were not observed for incorrect trials. As the authors note themselves, this differences between correct and incorrect trials might be due to the unequal number of correct (>90 percent of all trials) and incorrect trials. The authors offer bootstrapping analyses as control analysis, which is helpful and appreciated. However, while these control, analyses show that the reported findings can be observed in a relatively low number of correct trials, they do not rule out that very similar findings might be present for incorrect trials if there were more of these trials (in fact, even for the relatively few incorrect trials, there are trends). Thus, the conclusion of the authors should be downtoned in the sense that the effects appeared to be stronger for correct than for incorrect trials. Moreover, I thought that a more direct way to link neural activity patterns and behavioral performance would be via direct activity-performance correlations at the level of trials or groups of trials. Furthermore,

an inclusion of the different set sizes (which should result in different levels of performance) might be helpful here.

Reply: We agree that we cannot rule out the explanation that the lack of significant difference we observed is due to the small number of incorrect trials.

Action: Following the Reviewer's comment, we now removed the analyses on incorrect trials from the manuscript. Instead, we added analyses on WM load, as the number of trials with different loads is balanced.

In sum, the authors report several complex analyses and parameters, the interpretation of which is in my view at least less straightforward than suggested by the authors and would need further clarifications.

2.4 Amygdala during encoding and hippocampus during maintenance

Finally, the authors suggest on the one hand that the amygdala was preferentially involved at encoding (but see my comment above) and the hippocampus during maintenance and that there would be unidirectional information flow from the hippocampus to the amygdala during encoding and maintenance. I found these findings difficult to reconcile. If the amygdala contains stimulus-specific representations and is preferentially engaged during encoding, how should then the hippocampus inform the amygdala (and not vice versa). I think these findings would need a bit more of discussion.

Reply: We apologize for this misunderstanding here. Indeed, we observed bidirectional information flow between the amygdala and the hippocampus with a predominant information flow from the hippocampus to the amygdala, both during the encoding and the maintenance period. The observed information flow from the amygdala during the encoding period allows it to inform the hippocampus.

Besides, WM load could be decoded by using information flow from the amygdala during encoding (**Fig. 5(C)** left) and by using information flow from the hippocampus during maintenance (**Fig. 5(C)** right). This again indicated that the information flow driven by the amygdala during encoding and that driven by the hippocampus contributed to WM, and is in line with our previous findings of functional specialization.

Action: Following the Reviewer's suggestion, we have now extended the explanation of information flow between the two regions to the Discussion at pages 16-17:

In addition to the above-mentioned functional specialization, we found bidirectional inter-regional interaction between the amygdala and the hippocampus during WM encoding and maintenance. The inter-regional communication is consistent with rodent ³³ and monkey ³⁴ studies, which reported anatomical connection by using tracing techniques, and with rodent studies, which found synaptic plasticity in the amygdala induced by electrical stimulation to the hippocampus ³¹, and vice versa ³⁵. Human iEEG studies found inter-regional functional connectivity during emotional information ¹² and emotional memory ³² processing. In the current study we extended the functional interaction in the amygdala-hippocampal circuit to WM processing even in the absence of explicit emotional content. Besides, we found that the information flow from the amygdala contributes to WM encoding and that the information flow from the hippocampus contributes to WM maintenance in a load-sensitive manner. This is in line with previous studies which suggested that the inflow to the hippocampus during the encoding period transfers external sensory information, while the outflow from the hippocampus during the maintenance period transfers memory information ³⁶.

Reference

1. Boran E, *et al.* Persistent hippocampal neural firing and hippocampal-cortical coupling predict verbal working memory load. *Sci Adv* **5**, eaav3687 (2019).
2. Chang C-C, Lin C-J. LIBSVM: a library for support vector machines. *ACM transactions on intelligent systems and technology (TIST)* **2**, 1-27 (2011).
3. Mamashli F, *et al.* Synchronization patterns reveal neuronal coding of working memory content. *Cell Rep* **36**, 109566 (2021).
4. Oosterhof NN, Connolly AC, Haxby JV. CoSMoMVPA: Multi-Modal Multivariate Pattern Analysis of Neuroimaging Data in Matlab/GNU Octave. *Front Neuroinform* **10**, 27 (2016).
5. Johnson EL, *et al.* Dynamic frontotemporal systems process space and time in working memory. *PLoS Biol* **16**, e2004274 (2018).
6. Li J, *et al.* Anterior-Posterior Hippocampal Dynamics Support Working Memory Processing. *J Neurosci* **42**, 443-453 (2022).
7. Pacheco Estefan D, *et al.* Coordinated representational reinstatement in the human hippocampus and lateral temporal cortex during episodic memory retrieval. *Nat Commun* **10**, 2255 (2019).
8. Xiao X, Dong Q, Gao J, Men W, Poldrack RA, Xue G. Transformed neural pattern reinstatement during episodic memory retrieval. *J Neurosci* **37**, 2986-2998 (2017).
9. Goldman-Rakic PS. Cellular basis of working memory. *Neuron* **14**, 477-485 (1995).
10. Kaminski J, Brzezicka A, Mamelak AN, Rutishauser U. Combined Phase-Rate Coding by Persistently Active Neurons as a Mechanism for Maintaining Multiple Items in Working Memory in Humans. *Neuron* **106**, 256-264 e253 (2020).
11. Liu J, *et al.* Stable maintenance of multiple representational formats in human visual short-term memory. *Proc Natl Acad Sci U S A* **117**, 32329-32339 (2020).
12. Zheng J, *et al.* Amygdala-hippocampal dynamics during salient information processing. *Nat Commun* **8**, 14413 (2017).
13. Baddeley A. Working memory: theories, models, and controversies. *Annu Rev Psychol* **63**, 1-29 (2012).
14. Christophel TB, Klink PC, Spitzer B, Roelfsema PR, Haynes JD. The Distributed Nature of Working Memory. *Trends Cogn Sci* **21**, 111-124 (2017).
15. LaBar KS, Cabeza R. Cognitive neuroscience of emotional memory. *Nat Rev Neurosci* **7**, 54-64 (2006).
16. Gothard KM. Multidimensional processing in the amygdala. *Nat Rev Neurosci* **21**, 565-575 (2020).
17. Inman CS, *et al.* Direct electrical stimulation of the amygdala enhances declarative memory in humans. *Proc Natl Acad Sci U S A* **115**, 98-103 (2018).
18. Eichenbaum H. *Memory, amnesia, and the hippocampal system*. MIT press (1993).
19. Rutishauser U, Reddy L, Mormann F, Sarnthein J. The Architecture of Human Memory: Insights from Human Single-Neuron Recordings. *J Neurosci* **41**, 883-890 (2021).

20. Kornblith S, Quiñan Quiroga R, Koch C, Fried I, Mormann F. Persistent Single-Neuron Activity during Working Memory in the Human Medial Temporal Lobe. *Curr Biol* **27**, 1026-1032 (2017).
21. Kaminski J, Sullivan S, Chung JM, Ross IB, Mamelak AN, Rutishauser U. Persistently active neurons in human medial frontal and medial temporal lobe support working memory. *Nat Neurosci* **20**, 590-601 (2017).
22. Brzezicka A, Kaminski J, Reed CM, Chung JM, Mamelak AN, Rutishauser U. Working Memory Load-related Theta Power Decreases in Dorsolateral Prefrontal Cortex Predict Individual Differences in Performance. *J Cogn Neurosci* **31**, 1290-1307 (2019).
23. Freund MC, Etzel JA, Braver TS. Neural Coding of Cognitive Control: The Representational Similarity Analysis Approach. *Trends Cogn Sci* **25**, 622-638 (2021).
24. Favila SE, Chanales AJ, Kuhl BA. Experience-dependent hippocampal pattern differentiation prevents interference during subsequent learning. *Nat Commun* **7**, 11066 (2016).
25. Sander D, Grafman J, Zalla T. The human amygdala: an evolved system for relevance detection. *Rev Neurosci* **14**, 303-316 (2003).
26. Aggleton JP. *The amygdala: neurobiological aspects of emotion, memory, and mental dysfunction*. Wiley-Liss (1992).
27. Ranganath C, Ritchey M. Two cortical systems for memory-guided behaviour. *Nature Reviews Neuroscience* **13**, 713-726 (2012).
28. Vanz F, Bicca MA, Linartevichi VF, Giachero M, Bertoglio LJ, Monteiro de Lima TC. Role of dorsal hippocampus kappa opioid receptors in contextual aversive memory consolidation in rats. *Neuropharmacology* **135**, 253-267 (2018).
29. Zhang H, *et al*. Awake ripples enhance emotional memory encoding in the human brain. 2021.2011.2017.469047 (2021).
30. McDonald AJ, Mott DD. Functional neuroanatomy of amygdalohippocampal interconnections and their role in learning and memory. *J Neurosci Res* **95**, 797-820 (2017).
31. Maren S, Fanselow MS. Synaptic plasticity in the basolateral amygdala induced by hippocampal formation stimulation in vivo. *J Neurosci* **15**, 7548-7564 (1995).
32. Zheng J, *et al*. Multiplexing of Theta and Alpha Rhythms in the Amygdala-Hippocampal Circuit Supports Pattern Separation of Emotional Information. *Neuron* **102**, 887-898 e885 (2019).
33. Kishi T, Tsumori T, Yokota S, Yasui Y. Topographical projection from the hippocampal formation to the amygdala: a combined anterograde and retrograde tracing study in the rat. *J Comp Neurol* **496**, 349-368 (2006).
34. Amaral DG, Cowan WM. Subcortical afferents to the hippocampal formation in the monkey. *J Comp Neurol* **189**, 573-591 (1980).
35. Abe K, Niikura Y, Misawa M. The induction of long-term potentiation at amygdalo-hippocampal synapses in vivo. *Biol Pharm Bull* **26**, 1560-1562 (2003).
36. Dimakopoulos V, Mégevand P, Stieglitz LH, Imbach L, Sarnthein J. Information flows from hippocampus to auditory cortex during replay of verbal working memory items. *eLife* **11**, e78677 (2022).

REVIEWER COMMENTS

Reviewer #1 (Remarks to the Author):

The authors present the results of 14 neurosurgical patients with intracranial electrodes performing a working memory task. The main findings are that the amygdala exhibits greater differences among memory items following initial item presentation (EED), but the hippocampus exhibit greater similarity between encoding and maintenance period in a WM task. (EMS) I commend the authors on a novel paradigm asking an interesting question here with methods (EED esp) that are relatively innovative. The manuscript is exceptionally clear in its description of methods and results and is enjoyable to read.

The principal weakness of the first version of the manuscript was a lack of functional effects for the EMS/EED patterns the authors reported, making conclusions regarding connection to a WM process harder to support. To address this issue, they now include an SVM decoder analysis to look for an association between EED, EMS, or GC and the number of items (4, 6 or 8) which the individual was required to maintain. The results showed that during the encoding period, EED in the amygdala but not the hippocampus achieved above—chance classification while EMS revealed above chance classification for the hippocampus but for the amygdala. The same decoder style analysis was then used to analyze amygdala/hipp directionality using GC measures, suggesting that WM load could be decoded from different directional information during encoding versus maintenance periods.

I am not sure the author's conclusions regarding MTL circuitry in WM can be justified based on these SVM results. First, the decoder uses all available (trial level) data in a single model, which represents a substantial expansion of degrees of freedom in the analysis, making it less likely these results are generalizable. I do not think this improves the impact of the analysis as argued. Also, the validation method doesn't use a traditional training/testing set method (70/30 split for example) but rather trains on N-1 participants and then tests on N participants. Further, decoder accuracy was above chance but not by much, and the comparisons between regions for EED and EMS and for directionality rely on one result meeting the significance threshold while the other does not, but do not include a direct statistical comparison between the two distributions. Perhaps most important, the application of PCA to the input data before running through the SVM may create the possibility for the decoder to "peek" at the testing set, making the classification accuracy less reliable. If I misinterpreted these methodological steps in the execution of the SVM, then I apologize but this is my take away from the Methods description. The conclusions regarding the functional relevance of the EED and EMS patterns (related to load) would be more convincing with a regular random effects analysis.

These issues with the methods of analysis would be less of a problem for the underlying conclusions if this were a subsidiary analysis with a highline result predicated on more direct measures of a functional difference. But as reflected in the first round of review, a functional link is critical to interpret the

results. The authors removed the analysis related to incorrect trials, which is ok, but it means that conclusions related to WM effects in these circuits rely more heavily on the analysis of WM load. This matters for the control analysis related to differences in presence/absence of visual information (lack of a functional effect). The lack of a significant difference in ERPs is not sufficient by itself to control for this problem, as there are apparent differences in the average ERPs and the differences on a subject by subject basis may contribute to the classification results without being apparent in aggregate data. The best way to account for encoding/maintenance differences due to differences in visual information is to demonstrate a convincing effect related to a behavior, ie WM load or success effects.

The use of bipolar referencing for the GC analysis is a good addition for which the authors deserve credit. The application of Granger causality to ecog signals is challenging, since the method is designed for longer, relatively stable time series rather than trial by trial data which shows higher variance. Are these signals sufficiently stable for GC? Interpretation of the results is now a little more confusing, since the hipp->amy GC was higher across all frequencies in both encoding and maintenance but the amy->hipp direction was most significant in the decoder/functional analysis. An alternative approach such as phase slope index or mutual information might be less affected by signal stability.

A relatively minor point, but Figure 2G is misleading since they applied the t test to distributions of filtered by pixels that showed a significant difference in the previous step. Showing the bar plots for visualization is ok but the statistical result is not meaningful after such filtering.

Finally, I appreciate the change in terminology for the maintenance/encoding similarity analysis. The methodological visualization in the figures is good and makes sense. The electrode localization depictions would benefit from greater detail. Were all hippocampal contacts aggregated, ie anterior and posterior locations? The behavioral variance introduced by WM load suggests that success effects could perhaps be analyzed for the high load conditions in which performance was not at ceiling.

Reviewer #2 (Remarks to the Author):

The authors responded appropriately to all of my previous comments. I have no further comments.

Reviewer comments

We thank the expert Reviewer for the persistent engagement to improve our manuscript. The encouraging and constructive comments certainly helped us with our manuscript. Here, we provide a point-by-point response to all comments. Our responses are in blue, with text that was in the original submission in blue, and new text in red.

1 Reviewer #1:

1.1 General assessment

The authors present the results of 14 neurosurgical patients with intracranial electrodes performing a working memory task. The main findings are that the amygdala exhibits greater differences among memory items following initial item presentation (EED), but the hippocampus exhibit greater similarity between encoding and maintenance period in a WM task. (EMS) I commend the authors on a novel paradigm asking an interesting question here with methods (EED esp) that are relatively innovative. The manuscript is exceptionally clear in its description of methods and results and is enjoyable to read.

Reply: We thank the reviewer for this appreciation.

1.2 Changes in Revision 1

The principal weakness of the first version of the manuscript was a lack of functional effects for the EMS/EED patterns the authors reported, making conclusions regarding connection to a WM process harder to support. To address this issue, they now include an SVM decoder analysis to look for an association between EED, EMS, or GC and the number of items (4, 6 or 8) which the individual was required to maintain. The results showed that during the encoding period, EED in the amygdala but not the hippocampus achieved above-chance classification while EMS revealed above chance classification for the hippocampus but for the amygdala. The same decoder style analysis was then used to analyze amygdala/hipp directionality using GC measures, suggesting that WM load could be decoded from different directional information during encoding versus maintenance periods.

Reply: We agree with the Reviewer's summary of the changes our Revision 1.

1.3 Validity of SVM results

I am not sure the author's conclusions regarding MTL circuitry in WM can be justified based on these SVM results.

1.3.1 SVM 70/30 split and comparison between two distributions

First, the decoder uses all available (trial level) data in a single model, which represents a substantial expansion of degrees of freedom in the analysis, making it less likely these results are generalizable. I do not think this improves the impact of the analysis as argued. Also, the validation method doesn't use a traditional training/testing set method (70/30 split for example) but rather trains on N-1 participants and then tests on N participants. Further, decoder accuracy was above chance but not by much, and the comparisons between regions for EED and EMS and for directionality rely on one result meeting the significance threshold while the other does not, but do not include a direct statistical comparison between the two distributions.

Reply: We apologize for this misunderstanding and now explain our method more clearly. In our study, we applied leave-one-out cross-validation (LOOCV) at participant level. Specifically, the dataset from N participants was separated into a training data set from N-1 participants as well as a testing data set from the remaining one participant, not all N participants. Then the SVM classifier was trained by the training data set and tested by the testing data set. This procedure was replicated by N times. Although the LOOCV method are proposed to offer a relatively high detection power^{1, 2, 3}, we agree that a traditional training/testing set method (70/30 split for example) is more common method in machine learning analysis. We also agree that comparison between the two distributions is more direct than comparisons between the two distributions with the significance threshold respectively.

Action: Following the Reviewer's comments, we have now used a traditional training/testing set method (70/30 split here) and a direct statistical comparison between the two distributions. This analysis pipeline is now outlined in **Panel d in Fig. 1.**

Fig. 1 d Decoding analysis: we used machine-learning analyses to investigate whether WM load (load 4, 6 and 8) could be predicted by encoding-encoding dissimilarity (EED), encoding-maintenance similarity (EMS), or phase slope index (PSI) features.

We describe these new findings in the Results section:

Functional specialization and interaction within the amygdala-hippocampal circuit predicted WM load

Then, to address whether the EED patterns within the amygdala or the hippocampus could predict WM load, we developed an approach that uses the EED patterns in the amygdala as well as the hippocampus to predict the WM load (load 4, 6 or 8) with a linear support vector machine (SVM)⁴ classifier. For each load, the EED patterns at trial-pairs level for all participants were pooled as the data used in the classification. Then, we randomly extracted 70% of the data from each load and pooled them across all loads to train the SVM classifier. Next, the classifier was tested in the remaining data to get a decoding accuracy as performance measure. Then we did cross validation by permuting the whole data 100 times and each time replicating the classification process (see details in **Methods**). Then, we made direct comparison on the distribution of decoding accuracy between the amygdala and hippocampus using paired t-test. As shown in **Fig. 2h**, the decoding accuracy from the amygdala EED pattern ($33.54\% \pm 1.31\%$) was significantly higher than the hippocampus EED pattern ($32.96\% \pm 1.16\%$, $p = 0.0012$, $t(99) = 3.35$). We also performed analogical decoding analysis using the EMS patterns within the amygdala or the hippocampus, as described in the EED patterns. Results showed that the decoding accuracy ($35.33\% \pm 1.70\%$) from the hippocampus EMS pattern was significantly higher than the amygdala EMS pattern ($34.23\% \pm 1.50\%$, paired t -test: $p < 0.001$, $t(99) = 4.74$, **Fig. 3d**).

We followed up by investigating whether the inter-regional interaction between the amygdala and the hippocampus could predict WM load. Based on previous observations, we separately extracted the directional connectivity from the hippocampus leads as well as the amygdala leads on the z-scored PSI at each channel pair for each participant. Then, for both directions, the directional connectivity at channel-pairs level for all participants were pooled as the data for each load classification (see details in **Methods**). Similar decoding analyses were performed as described above for each direction during encoding and maintenance, and we made direct comparisons on the decoding accuracy between both directions using paired t-test. As presented in **Fig. 4c**, during encoding, the decoding accuracy using the features from the amygdala leads ($42.94\% \pm 3.22\%$) was significantly higher than the opposite direction ($40.85\% \pm 3.08\%$; paired t -test: $p < 0.001$, $t(99) = 4.62$). While during maintenance, the decoding accuracy using the features from the hippocampus leads ($45.62\% \pm 3.57\%$) was higher than the opposite direction ($43.75\% \pm 3.21\%$; paired t -test: $p < 0.001$, $t(99) = 3.75$; **Fig. 4d**).

Fig. 2h When using the EED patterns, the decoding accuracy within the amygdala (red) is higher than in the hippocampus (blue). Dotted lines indicate the median. Broken lines above and below denote the quartiles. ** $p < 0.01$.

Fig. 3d The decoding accuracy by using the EMS patterns within the amygdala (red) and the hippocampus (blue). Dotted lines indicate the median. Broken lines above and below denote the quartiles. *** $p < 0.001$.

Fig. 4c-d The decoding accuracy by using the PSI features from the amygdala leads (red), from the hippocampus leads (blue) during encoding and maintenance periods. Dotted lines indicate the median. Broken lines above and below denote the quartiles. *** $p < 0.001$.

We now describe the new decoding analysis in the Methods section:

Decoding analysis

(A) EED pattern: We first considered the EED patterns within the amygdala and the hippocampus to decode the WM load. For each load, the EED pattern at trial-pairs level for all participants were merged as the data ($100 \text{ trial-pairs} \times 14 \text{ participants} = 1400 \text{ samples}$) used in the classification. The EED pattern included $201 \times 201 = 40401$ (201 denotes time windows during encoding) values, and these values were converted to a feature vector. Then, we split 70%

of the data from each load and merged them across all loads as the training data set. The remaining data were pooled across all loads as the testing data set. Meanwhile, to reduce the feature dimensionality, principal component analysis (PCA) was applied to the training data set to keep several principal components (K components) that explained 99% of the variance in the data. We also transformed the testing data set with the PCA matrix that was already fitted to the training data set. In total, we had the training data set with (980×3) samples $\times K$ features and the testing data set with (420×3) samples $\times K$ features. We trained an SVM classifier with a linear kernel with a cost equal to one. This procedure was replicated by 100 times for the cross-validation. Accuracy of the classifier as performance measures was averaged across 100 cross-validations. We performed this classification by using EED features in the amygdala and the hippocampus, separately.

(B) EMS pattern: Next, we decoded the WM load by using the EMS features within the amygdala and the hippocampus, respectively. Decoding analyses were performed as described for the EED patterns. The EMS patterns included $201 \times 301 = 60501$ values and were converted to a feature vector. Similar as described for EED patterns, for each load, we combined the EMS features at trial level for all participants as the data, then split 70% of the data as the training data set (N samples), then applied PCA to the training data set to K components that explained 99% of the variance, then fed the features $((N_{load4} + N_{load6} + N_{load8})$ samples $\times K$ features) into a linear SVM classifier for training, and tested the model on the remaining pooled data set that already applied K components matrix to the testing data set $((M_{load4} + M_{load6} + M_{load8})$ samples $\times K$ features). The accuracy was averaged across 100 cross-validations. Again, this classification was performed by using EMS features in the amygdala and the hippocampus, separately.

(C) PSI pattern: Then we tested whether WM load could be decoded by the directional connectivity from the hippocampus leads as well as the amygdala leads during encoding and maintenance, separately. We first extracted the directional connectivity from the hippocampus leads as well as the amygdala leads based on the z-scored PSI at each channel-pair for each participant. Then, the directional connectivity was vectorized (K features) at channel-pairs level for

all participants and was merged as the data for each load classification (172 channel-pairs). Again, we split the directional connectivity for each load using 70/30 split, with 70% of the data (120 samples) for training and the remaining data (52 samples) for testing. Then, we fed the training data set ((120 × 3) samples × K features) into the SVM classifier and tested it on the testing data set ((52 × 3) samples × K features). This procedure was replicated 100 times for cross-validation. In total, we performed four classifications using PSI features from the hippocampus leads and from the reverse direction during encoding and maintenance, separately.

For each decoding analysis, we made direct comparisons on the distribution of the decoding accuracy between two regions or directions using paired t -test. $P < 0.05$ was considered as significance.

1.3.2 Principal Component Analysis (PCA) is reliable

Perhaps most important, the application of PCA to the input data before running through the SVM may create the possibility for the decoder to “peek” at the testing set, making the classification accuracy less reliable. If I misinterpreted these methodological steps in the execution of the SVM, then I apologize but this is my take away from the Methods description.

Reply: We apologize for this misunderstanding and now explain our PCA approach more clearly. Indeed, we first applied PCA only on the training data set, and then we transformed the test data set using the already fitted PCA matrix. Thus, the decoder trained by the training set did not “peek” at the testing set.

Action: We revised the descriptions on how we applied PCA to the training and testing datasets in the Methods section:

Decoding analysis

Meanwhile, to reduce the feature dimensionality, principal component analysis (PCA) was applied to the training data set to keep several principal components (K components) that explained 99% of the variance in the data. We also transformed the testing data set with the PCA matrix that was already fitted to the training data set.

1.3.3 Random Effects Analysis

The conclusions regarding the functional relevance of the EED and EMS patterns (related to load) would be more convincing with a regular random effects analysis.

Reply: We thank the Reviewer for this suggestion.

Action: Following up the Reviewer's comment, we made the random effects analysis to compare the EED/EMS patterns between WM load respectively. We treated the regions (amygdala / hippocampus) and WM load (low (set4) / high (set6/8)) as fixed factors, the participants as random factor, and the extracted EED/EMS values as the dependent variables. We speculated that EED values increased in high-load trials that have more number of letters and thus more variation between trials. We also expected decreased EMS values in high-load trials as high-load trials with more letters are more difficult to maintain and result in lower similarity between encoding and maintenance periods.

As expected, we observed higher EED values and lower EMS values for high-load trials versus low-load trials regardless of the regions, although the load effect did not reach significance (see supplementary **Fig. S2c-d**). Despite the lack of significant effect identified by univariate analyses, the classifier using multivariate analyses showed significant difference in decoding accuracy between the amygdala and hippocampus. Specifically, the decoding accuracy from the amygdala EED pattern ($33.54\% \pm 1.31\%$) was significantly higher than the hippocampus EED pattern ($32.96\% \pm 1.16\%$, $p = 0.0012$, $t(99) = 3.35$, **Fig. 2h**). The decoding accuracy ($35.33\% \pm 1.70\%$) from the hippocampus EMS pattern was significantly higher than the amygdala EMS pattern ($34.23\% \pm 1.50\%$, paired t-test: $p < 0.001$, $t(99) = 4.74$, **Fig. 3d**) (also see Reply 1.3.1). Multivariate pattern analysis (MVPA) allows the detection of aggregated weak distributed effects and is more informative and sensitive than univariate analysis⁵. These results together indicate that the EED pattern from the amygdala and the EMS pattern from the hippocampus predict WM load, respectively.

We describe these new findings in the Results section:

Functional specialization and interaction within the amygdala-hippocampal circuit predicted WM load

In addition, we made a random effects analysis to directly compare the EED patterns between WM load. We treated the regions (amygdala / hippocampus) and WM load (low (set4) / high (set6/8)) as fixed factors, the participants as random factor, and the extracted EED values as the dependent variables. We observed higher EED values for high-load trials versus low-load trials regardless of the regions, although the load effect did not reach significance (mixed-effect model: $p = 0.14$; **Fig. S2c**). Similar comparison was also made between WM load as described in EED patterns. As shown in **Fig. S2d**, lower EMS values with high-load trials were obtained relative to the low-load trials regardless of the regions, although the load effect did not reach significance (mixed-effect model: $p = 0.25$).

Supplementary Figure S2c-d. EED and EMS values in high-load trials (dark color) and low-load trials (light color) in the amygdala (red) and hippocampus (blue). No difference was found between high-load and low-load trials.

1.4 Inter-regional differences in visual processing

These issues with the methods of analysis would be less of a problem for the underlying conclusions if this were a subsidiary analysis with a highline result predicated on more direct measures of a functional difference. But as reflected in the first round of review, a functional link is critical to interpret the results. The authors removed the analysis related to incorrect trials, which is ok, but it means that conclusions related to WM effects in these circuits rely more heavily on the analysis of WM load. This matters for the control analysis related to differences in presence/absence of visual information (lack of a functional effect). The lack of a significant difference in ERPs is not sufficient by itself to control for this problem, as there are apparent differences in the average ERPs and the differences on a participant-by-participant basis may contribute to the classification results without

being apparent in aggregate data. The best way to account for encoding/maintenance differences due to differences in visual information is to demonstrate a convincing effect related to a behavior, ie WM load or success effects.

Reply: We thank the Reviewer for this suggestion. We agree that we should exclude the possibility that our results are due to differences in visual processing to ensure our finding of regional difference in encoding (or maintenance) processing is convincing.

Action: Given the presence of visual information during the encoding period and the absence during the maintenance period, we calculated the difference of power between the two periods ((maintenance-encoding)/encoding) to index the processing of visual information in the amygdala and the hippocampus. To test whether there is regional difference in visual processing, we first compared the visual processing index between the amygdala and the hippocampus by using paired t-test, then we compared the decoding accuracy of WM load between using this index in two regions.

These results are now presented in the Results section:

Functional specification: Stable representation within hippocampus during maintenance

We next exclude the possibility that the inter-regional differences were due to the presence of visual information during encoding and its absence during maintenance. To this end, we calculated the difference of power between the two periods ((maintenance-encoding)/encoding) to index the processing of visual information in the amygdala and the hippocampus and made comparison between regions using paired t-tests. No difference was found between the two regions (paired *t*-test, $p = 0.09$, **Fig. S2e**). Among all 14 participants, the relative difference of power was higher within the amygdala in 7 participants and higher within the hippocampus in 7 participants. Further, we also extracted this relative difference of power to decode the WM load. As shown in **Fig. S2f**, the relative difference of power could not decode the WM load regardless of using features within the amygdala ($32.17\% \pm 10.79\%$) or within the hippocampus ($32.58\% \pm 8.13\%$; permutation test against scrambled data, $p > 0.05$). No difference of decoding accuracy was found between the two regions (paired *t*-test, $p = 0.76$).

Supplementary Figure S2 e Relative difference of power indicating encoding/maintenance difference within the amygdala (red) and the hippocampus (blue) for each participant. No difference was found between regions. f Decoding accuracy using the relative difference of power within the amygdala (red) and the hippocampus (blue). WM load could not be decoded by features from either region and no difference was found between regions. Dotted lines indicate the median. Broken lines above and below denote the quartiles.

1.5 Validity of Granger Causality analysis (GC)

The use of bipolar referencing for the GC analysis is a good addition for which the authors deserve credit. The application of Granger causality to ecog signals is challenging, since the method is designed for longer, relatively stable time series rather than trial by trial data which shows higher variance. Are these signals sufficiently stable for GC? Interpretation of the results is now a little more confusing, since the hipp->amy GC was higher across all frequencies in both encoding and maintenance but the amy->hipp direction was most significant in the decoder/functional analysis. An alternative approach such as phase slope index or mutual information might be less affected by signal stability.

Reply: We know from GC analysis on related data for the same task that the signals are sufficiently stable⁶. Still, we thank the reviewer for recommending the phase slope index (PSI) to corroborate our results.

Action: Following the Reviewer' suggestion, we have now computed the PSI during the encoding and the maintenance periods, and then analyzed how it varied with WM load. We now describe the new PSI findings in the Results section:

Functional interaction: Bidirectional information transfer within the amygdala-hippocampal circuit during WM encoding and maintenance

...We used Phase Slope Index (PSI) to quantify the directional connectivity between the amygdala and the hippocampus⁷. PSI quantifies phase difference as a function of frequency, with a positive value indicating that the signal from the first structure is leading the signal from the second structure. PSI was computed for the data segments during encoding and maintenance for all correct trials from 1 to 40 Hz and tested for significance of directional effects via a nonparametric permutation procedure (see **Methods** for details). The directional effects were averaged across channel pairs and participants, yielding a z-score that indicates the information flow between the amygdala and the hippocampus during encoding and maintenance, respectively, as previous studies did⁸. The encoding period was characterized by unidirectional hippocampus-to-amygdala connectivity across 1-11 Hz (threshold $z > 1.96$, $p < 0.05$), and unidirectional amygdala-to-hippocampus connectivity across 13-27 Hz and 36-40 Hz (threshold $z < -1.96$, $p < 0.05$, **Fig. 4a**). The maintenance period was characterized by unidirectional hippocampus-to-amygdala connectivity across 1-18 Hz (threshold $z > 1.96$, $p < 0.05$), and unidirectional amygdala-to-hippocampus connectivity across 23-30 Hz and 33-34 Hz (threshold $z < -1.96$, $p < 0.05$, **Fig. 4b**). These results indicated a frequency-specific directional connectivity in the amygdala-hippocampal circuit involved in WM processing. Specifically, the direction of influence differed across frequency band, with theta/alpha-driven unidirectional influence from the hippocampus and beta-driven influence from the amygdala. Besides, frequency bands for both directions varied between the encoding and the maintenance period. This resulted in a wider frequency band range showing amygdala leads influence than the opposite direction in the encoding period, and a wider frequency with hippocampal leads influence in the maintenance period. These findings are consistent with findings in the representational analyses showing contribution of the amygdala to WM encoding and the hippocampus to WM maintenance.

Functional specialization and interaction within the amygdala-hippocampal circuit predicted WM load

We followed up by investigating whether the inter-regional interaction between the amygdala and the hippocampus could predict WM load. Based on previous

observations, we separately extracted the directionality pattern from the hippocampus leads as well as the amygdala leads on the z-scored PSI at each channel pair for each participant. Then, for both directions, the directionality pattern at channel-pairs level for all participants were merged as the data for each load classification (see details in **Methods**). Similar decoding analyses were performed as described above for each direction during encoding and maintenance separately, and we made direct comparisons on the decoding accuracy between both directions using paired *t*-test. As presented in **Fig. 4c**, during encoding, the decoding accuracy using the features from the amygdala leads ($42.94\% \pm 3.22\%$) was significantly higher than the opposite direction ($40.85\% \pm 3.08\%$; paired *t*-test: $p < 0.001$, $t(99) = 4.62$). While during maintenance, the decoding accuracy using the features from the hippocampus leads ($45.62\% \pm 3.57\%$) was higher than the opposite direction ($43.75\% \pm 3.21\%$; paired *t*-test: $p < 0.001$, $t(99) = 3.75$; **Fig. 4d**).

New Fig. 4 Directional information flow between the hippocampus and the amygdala during encoding and maintenance. a The z-scored PSI across 1-40 Hz during encoding. Asterisks in blue denote significant PSI from the hippocampus to the amygdala and these in red denote the opposite direction (significance was thresholded at $|z| > 1.96$). **b** The z-scored PSI across 1-40 Hz during maintenance. **c** Decoding accuracy of WM load by using PSI features of the amygdala leads connectivity (red) was higher than those of the hippocampus leads connectivity

(blue) during encoding. Dotted lines indicate the median. Broken lines above and below denote the quartiles. *** $p < 0.001$. **d** Decoding accuracy of WM load by using PSI features of the hippocampus leads connectivity (blue) was higher than those of the amygdala leads connectivity (red) during maintenance. *** $p < 0.001$.

And the PSI analysis in the Methods section:

Phase slope index analysis

PSI estimates whether the slope of the phase differences between A-B signal pairs is consistent across several adjacent frequency bins, in which positive PSI indicates that signal A leads signal B, and negative PSI indicates the reverse⁹. In this study, the data segments during encoding and maintenance were zero padded and multiplied with a Hann taper from 1 to 40 Hz with 1 Hz step, from which we computed the PSI at each inter-regional channel pair within the same hemisphere in each participant (i.e., one channel from the anterior/posterior hippocampus and the other from the amygdala) and pooled all possible channel pairs between the hippocampus and the amygdala for each participant. To correct for any spurious results, we randomly shuffled the trials and recomputed the PSI at each channel pair. This step was repeated 200 times to create normal distributions of channel pair-resolved null PSI data. To construct a directional effect of the amygdala-hippocampus on a population level, we averaged the raw PSI across channel pairs and participants. Correspondingly, the null distributions were also averaged across channel pairs and participants. Consequently, the raw PSI outputs can be compared to the distribution of null PSI to derive a z-score in the frequency band 1-40 Hz (for a similar approach, see Solomon et al.⁸). Significant PSI was thresholded at $|z| > 1.96$, in which the hippocampus leads were defined as $z > 1.96$ and the amygdala leads as $z < -1.96$, as previous studies performed^{9,10}.

1.6 Figure 2G

A relatively minor point, but Figure 2G is misleading since they applied the t test to distributions of filtered by pixels that showed a significant difference in the previous step. Showing the bar plots for visualization is ok but the statistical result is not meaningful after such filtering.

Reply: We thank the Reviewer for the points regarding the figure.

Action: We have now adapted Figure 2g changed the caption accordingly.

Fig. 2g. EED values averaged over the significant cluster in **b** was extracted within the hippocampus (blue) and amygdala (red) for each participant, respectively. 12 of 14 participants showed higher EED values within the amygdala than within the hippocampus.

1.7 Terminology

Finally, I appreciate the change in terminology for the maintenance/encoding similarity analysis. The methodological visualization in the figures is good and makes sense.

Reply: We thank the Reviewer for this appreciation.

1.8 Electrode localization

The electrode localization depictions would benefit from greater detail. Were all hippocampal contacts aggregated, ie anterior and posterior locations?

Reply: We apologize for the missing information in the original manuscript. We selected the two most medial channels on each electrode targeting the anterior or posterior hippocampus, so that the number of channels in the anterior and posterior hippocampus are balanced. The power was then computed for each selected hippocampal channel in each participant. Next, when we calculated the EED or EMS within the hippocampus, the power pattern across channels and frequencies within each time window were vectorized, as a recent study did ¹¹. We computed the PSI between signals at each inter-regional channel pair within the same hemisphere (e.g., one from the anterior/posterior hippocampus and the other from the amygdala),

and then we pooled the PSI across all possible channel pairs between the hippocampus and the amygdala for each participant.

Action: We have now updated the descriptions of aggregation of hippocampal channels in the Methods section:

Time-frequency analysis

Time-frequency power was computed for each selected channel at each trial within the hippocampus as well as the amygdala in each participant.

Phase slope index analysis

...In this study, the data segments during encoding and maintenance were zero padded and multiplied with a Hann taper from 1 to 40 Hz with 1 Hz step, from which we computed the PSI at each inter-regional channel pair within the same hemisphere in each participant (i.e., one channel from the anterior/posterior hippocampus and the other from the amygdala) and pooled all possible channel pairs between the hippocampus and the amygdala for each participant.

1.9 Success effect for high load conditions

The behavioral variance introduced by WM load suggests that success effects could perhaps be analyzed for the high load conditions in which performance was not at ceiling.

Reply: We thank the Reviewer for this suggestion.

Action: Following the Reviewer's suggestion, we have now analyzed the success on trials with high load. We describe these findings in the Results section:

Success effect for high load conditions

In addition, we examined whether the neural representation and directional connectivity varied as a function of success, as the accuracy in the high load conditions (combined load 6 and 8) is not at ceiling (mean performance over all participants 87%). First, we separately computed the EED and EMS for the correct and incorrect trials in the high load conditions. We then performed two of 2 (Performance: correct vs. incorrect) × 2 (Region: amygdala vs. hippocampus) repeated-measures ANOVAs, one with the EED value and one with the EMS value as the dependent variable. As shown in **Fig. S3a**, the EED value of the correct trials were significantly greater than incorrect trials ($p = 0.026$, $F(1,13) = 6.28$). This

indicated that the activity patterns for correct trials showed a larger distance among different trials, whereas incorrect trials showed overlapping representations across different items with reduced neural dissimilarity. The EMS showed no significant difference between the correct trials and incorrect trials ($p = 0.17$). Regarding to the PSI, for each participant, we separately extracted the PSI from the “hippocampus leads” and the “amygdala leads” for the correct and incorrect trials. Again, we made comparisons between regions and performance by using repeated-measures ANOVAs. During encoding, we found a significant interaction effect ($p = 0.019$, $F(1,13) = 7.23$). Further analysis showed that the amygdala leads connectivity was significantly larger in the correct trials than the incorrect trials ($p = 0.042$) and no difference between correct and incorrect trials was found from the opposite direction ($p = 0.21$, **Fig. S3b**). During maintenance, a significant interaction effect ($p = 0.025$, $F(1,13) = 6.43$) was also found. Further analysis showed that the hippocampus leads connectivity was significantly larger in the correct trials than the incorrect trials ($p = 0.034$) and no difference was found between correct and incorrect trials from the opposite direction ($p = 0.14$, **Fig. S3c**). This again indicated that the information flow driven by the amygdala during encoding and that driven by the hippocampus contributed to WM.

Next, we also applied the EED/EMS/PSI patterns within the amygdala and the hippocampus to decode the performance (correct or incorrect) in the high load conditions. Using the SVM classifier as described before, we found that the decoding accuracy by using the EED pattern within the amygdala ($61.50\% \pm 13.73\%$) was higher than the hippocampus ($50.25\% \pm 10.66\%$; paired t -test, $p < 0.001$, $t(99)=6.85$; **Fig. S3d**). The decoding accuracy by using the EMS pattern within the hippocampus ($59.63\% \pm 12.29\%$) was higher than the amygdala ($56.13\% \pm 10.88\%$; paired t -test, $p = 0.030$, $t(99)=2.20$; **Fig. S3e**). Besides, during encoding, decoding accuracy by using the “amygdala leads” PSI ($59.50\% \pm 17.69\%$) was higher than the “hippocampus leads” PSI ($53.88\% \pm 14.40\%$; paired t -test, $p = 0.022$, $t(99)=2.33$; **Fig. S3f**); and during maintenance, the decoding accuracy by using the “hippocampus leads” PSI ($61.75\% \pm 11.35\%$) was higher than that the “amygdala leads” PSI ($57.88\% \pm 14.40\%$; paired t -test, $p = 0.016$, $t(99)=2.46$; **Fig. S3g**). Taken together, these results indicate in the high load conditions that the contribution of EED within the amygdala and amygdala leads directional connectivity on WM

performance during encoding, and the contribution of EMS within the hippocampus and hippocampus leads directional connectivity on WM performance during maintenance.

Supplementary Fig. S3. Success effects for the high load conditions and corresponding decoding accuracy within the amygdala and the hippocampus.

a The EED values within the amygdala (red) and the hippocampus (blue) for the correct (left) and incorrect trials (right), respectively. The EED values for the correct trials were significantly higher than those for the incorrect trials. * $p < 0.05$. **b-c** The PSI directionality of the amygdala leads connectivity (red) and the hippocampal leads connectivity (blue) for the correct and incorrect trials during encoding (**b**) and maintenance (**c**), respectively. Positive values indicate the "hippocampus leads" connectivity, and negative values indicate the "amygdala leads" connectivity. During encoding, the amygdala leads connectivity was significantly larger at the correct trials than the incorrect trials while no difference was found from the opposite direction. During maintenance, the hippocampal leads connectivity was significantly larger at the correct trials than the incorrect trials while no difference was found from the opposite direction. * $p < 0.05$. **d** Decoding accuracy using the EED patterns within the amygdala (red) and the hippocampus (blue). Using the features from the amygdala was able to decode the WM performance with higher decoding accuracy

than that from the hippocampus. *** $p < 0.001$. **e** Decoding accuracy using the EMS patterns within the amygdala (red) and the hippocampus (blue). Using the features from the hippocampus was able to decode the WM performance with higher decoding accuracy than that from the amygdala. * $p < 0.05$. **f-g** Decoding accuracy using the PSI features from both directions during encoding (**f**) and maintenance (**g**). WM performance could be better decoded with the PSI features from the amygdala leads during encoding and the PSI features from the hippocampus leads during maintenance. * $p < 0.05$.

References

1. Wang Q, Cagna B, Chaminade T, Takerkart S. Inter-subject pattern analysis: A straightforward and powerful scheme for group-level MVPA. *NeuroImage* **204**, 116205 (2020).
2. Wang Q, Artières T, Takerkart S. Inter-subject pattern analysis for multivariate group analysis of functional neuroimaging. A unifying formalization. *Computer Methods and Programs in Biomedicine* **197**, 105730 (2020).
3. Melbaum S, *et al.* Conserved structures of neural activity in sensorimotor cortex of freely moving rats allow cross-subject decoding. *Nature Communications* **13**, 7420 (2022).
4. Chang C-C, Lin C-J. LIBSVM: a library for support vector machines. *ACM transactions on intelligent systems and technology (TIST)* **2**, 1-27 (2011).
5. Norman KA, Polyn SM, Detre GJ, Haxby JV. Beyond mind-reading: multi-voxel pattern analysis of fMRI data. *Trends Cogn Sci* **10**, 424-430 (2006).
6. Dimakopoulos V, Mégevand P, Stieglitz LH, Imbach L, Sarnthein J. Information flows from hippocampus to auditory cortex during replay of verbal working memory items. *eLife* **11**, e78677 (2022).
7. Nolte G, *et al.* Robustly estimating the flow direction of information in complex physical systems. *Phys Rev Lett* **100**, 234101 (2008).
8. Solomon EA, *et al.* Dynamic Theta Networks in the Human Medial Temporal Lobe Support Episodic Memory. *Curr Biol* **29**, 1100-1111 e1104 (2019).
9. Johnson EL, *et al.* Dynamic frontotemporal systems process space and time in working memory. *PLoS Biol* **16**, e2004274 (2018).
10. Li J, *et al.* Anterior-Posterior Hippocampal Dynamics Support Working Memory Processing. *J Neurosci* **42**, 443-453 (2022).
11. Liu J, *et al.* Stable maintenance of multiple representational formats in human visual short-term memory. *Proc Natl Acad Sci U S A* **117**, 32329-32339 (2020).

REVIEWERS' COMMENTS

Reviewer #1 (Remarks to the Author):

I appreciate the authors' responsiveness and comprehensive work on this manuscript. PSI nicely augments the Granger analysis. Demonstrating the stability of the signal is reasonable also. The analysis of success effects in the high load condition is a nice addition. This is the most convincing result in my estimation and enhances the paper.

I have one question

Why are the degrees of freedom for the hipp/amygdala comparisons 99? Is it because they are testing across all of the permutations? The effect sizes are pretty small (less than 1% difference) and each of the permutations is not independent. You could update this analysis by shuffling the labels and generating a null distribution that includes both amygdala/hipp data and observing where the amygdala/hipp classifier difference falls within the distribution of classifier differences from the shuffle. But, unless I missed something, testing across all the permutations as independent observations does not seem valid.

Reviewer comments

Reviewer #1:

I appreciate the authors' responsiveness and comprehensive work on this manuscript. PSI nicely augments the Granger analysis. Demonstrating the stability of the signal is reasonable also. The analysis of success effects in the high load condition is a nice addition. This is the most convincing result in my estimation and enhances the paper.

Reply: We thank the reviewer for the thorough review and positive feedback on our revised manuscript. Our responses are in blue, with text that was in the original submission in blue, and new text in red.

I have one question:

Why are the degrees of freedom for the hipp/amygdala comparisons 99? Is it because they are testing across all of the permutations? The effect sizes are pretty small (less than 1% difference) and each of the permutations is not independent. You could update this analysis by shuffling the labels and generating a null distribution that includes both amygdala/hipp data and observing where the amygdala/hipp classifier difference falls within the distribution of classifier differences from the shuffle. But, unless I missed something, testing across all the permutations as independent observations does not seem valid.

Reply: We thank the Reviewer for suggesting permutation tests.

Action: Following the Reviewer's suggestion, we updated paired-t tests with permutation tests. Specifically, we shuffled the labels and compared the amygdala/hippocampus classifier difference to the null distribution of classifier differences resulting from label shuffling. Our findings remained valid with the permutation tests. Specifically, we found higher decoding accuracy for working memory (WM) load by using encoding-encoding dissimilarity (EED)

within the amygdala and amygdala-driven information flow during encoding, as well as higher decoding accuracy for WM load by using encoding-maintenance (EMS) within the hippocampus and hippocampus-driven information flow during maintenance.

We have now entered these updated tests in the Results section:

Functional specialization and interaction within the amygdala-hippocampal circuit predicted WM load

...We randomly extracted 70% of the data from each load and pooled them across all loads to train the SVM classifier. We tested the classifier in the remaining data to obtain the decoding accuracy as our performance measure.

The procedure was repeated 100 times for cross-validations (see details in Methods), and the accuracy of the classifier was averaged across these 100 cross-validations to measure its performance. The significance of the difference in decoding accuracy between the amygdala and hippocampus was assessed using a nonparametric permutation test. Specifically, we compared the actual difference with a null distribution obtained from scrambled labels. As shown in **Fig. 2h**, the decoding accuracy from the amygdala EED pattern ($33.54\% \pm 1.31\%$) was significantly higher than the hippocampus EED pattern ($32.96\% \pm 1.16\%$; permutation test: $p = 0.01$). We also performed analogical decoding analysis using the EMS patterns within the amygdala or the hippocampus, as described in the EED patterns. Results (**Fig. 3d**) showed that the decoding accuracy ($35.33\% \pm 1.70\%$) from the hippocampus EMS pattern was significantly higher than the amygdala EMS pattern ($34.23\% \pm 1.50\%$; permutation test: $p < 0.001$).

...As presented in **Fig. 4c**, during encoding, the decoding accuracy using the features from the amygdala leads ($42.94\% \pm 3.22\%$) was significantly higher than the opposite direction ($40.85\% \pm 3.08\%$; permutation test: $p < 0.001$). While during maintenance (**Fig. 4d**), the decoding accuracy using the features from the hippocampus leads ($45.62\% \pm 3.57\%$) was higher than the opposite direction ($43.75\% \pm 3.21\%$; permutation test: $p < 0.001$).

Success effect for high load conditions

...Using the SVM classifier as described before, we found that the decoding accuracy by using the EED pattern within the amygdala ($61.50\% \pm 13.73\%$) was higher than the hippocampus ($50.25\% \pm 10.66\%$; permutation test: $p < 0.001$; Fig. S3d). The decoding accuracy by using the EMS pattern within the hippocampus ($59.63\% \pm 12.29\%$) was higher than the amygdala ($56.13\% \pm 10.88\%$, although the difference did not reach significance (permutation test: $p = 0.062$; Fig. S3e). Besides, during encoding, decoding accuracy by using the “amygdala leads” PSI ($59.50\% \pm 17.69\%$) was higher than the “hippocampus leads” PSI ($53.88\% \pm 14.40\%$; permutation test: $p = 0.005$; Fig. S3f); and during maintenance, the decoding accuracy by using the “hippocampus leads” PSI ($61.75\% \pm 11.35\%$) was higher than that the “amygdala leads” PSI ($57.88\% \pm 14.40\%$; permutation test: $p = 0.025$; Fig. S3g).

We now describe the new decoding analysis in the Methods section:

Decoding analysis

For each decoding analysis, we used a nonparametric permutation test to evaluate the significance of the decoding accuracy difference between two regions or directions. Specifically, we shuffled the labels 200 times, and in each shuffling, we calculated decoding accuracy differences between two regions or directions, resulting in a null distribution that encompasses the data. P values were computed by comparing observed decoding accuracy difference with the entire distribution of null differences in decoding accuracy. We considered $P < 0.05$ to be significant.